# TAPBPR mediates peptide dissociation from MHC class I using a leucine lever

**F Tudor Ilca[1], Andreas Neerincx[1], Clemens Hermann[2], Ana Marcu[3], Stefan Stevanović[3,4], Janet E Deane[5], Louise H Boyle[1]\***

[1]Department of Pathology, University of Cambridge, Cambridge, United Kingdom; [2]Department of Integrative Biomedical Sciences, Division of Chemical and Systems Biology, Institute of Infectious Disease and Molecular Medicine, University of Cape Town, Cape Town, South Africa; [3]Department of Immunology, Interfaculty Institute for Cell Biology, University of Tübingen, Tübingen, Germany; [4]DKFZ Partner Site Tübingen, German Cancer Consortium, Tübingen, Germany; [5]Cambridge Institute for Medical Research, University of Cambridge, Cambridge, United Kingdom

**Abstract** Tapasin and TAPBPR are known to perform peptide editing on major histocompatibility complex class I (MHC I) molecules; however, the precise molecular mechanism(s) involved in this process remain largely enigmatic. Here, using immunopeptidomics in combination with novel cell-based assays that assess TAPBPR-mediated peptide exchange, we reveal a critical role for the K22-D35 loop of TAPBPR in mediating peptide exchange on MHC I. We identify a specific leucine within this loop that enables TAPBPR to facilitate peptide dissociation from MHC I. Moreover, we delineate the molecular features of the MHC I F pocket required for TAPBPR to promote peptide dissociation in a loop-dependent manner. These data reveal that chaperone-mediated peptide editing on MHC I can occur by different mechanisms dependent on the C-terminal residue that the MHC I accommodates in its F pocket and provide novel insights that may inform the therapeutic potential of TAPBPR manipulation to increase tumour immunogenicity.

DOI: https://doi.org/10.7554/eLife.40126.001

**\*For correspondence:**
lhb22@cam.ac.uk

## Introduction

Major histocompatibility complex class I (MHC I) molecules play a critical role in immmunosurveillance, particularly in the context of viral infections and cancer, by presenting antigenic peptides to CD8 +T cells. Prior to their cell surface export, MHC I molecules undergo peptide editing, a process that involves the exchange of low-affinity peptides for those of higher affinity. In addition to ensuring that only stable peptide:MHC I complexes are released to the plasma membrane, peptide editing ultimately controls the peptide repertoire that is displayed for immune detection. For over two decades, tapasin was the only known peptide editor for MHC I, facilitating peptide selection within the confines of the peptide loading complex (PLC) (*Williams et al., 2002*; *Howarth et al., 2004*; *Chen and Bouvier, 2007*; *Wearsch and Cresswell, 2007*). However, it is now well-recognised that the tapasin-related protein TAPBPR is a second independent peptide editor that performs peptide exchange outside the PLC (*Boyle et al., 2013*; *Hermann et al., 2013*; *Hermann et al., 2015*; *Morozov et al., 2016*). Furthermore, TAPBPR can work in cooperation with UDP-glucose:glycoprotein glucosyltransferase 1 (UGT1) to reglucosylate MHC I, causing recycling of MHC molecules to the PLC (*Neerincx et al., 2017*).

Although TAPBPR usually functions as an intracellular peptide editor, we have recently made the fascinating discovery that when given access to surface expressed MHC I molecules, TAPBPR retains its function as a peptide exchange catalyst and can be utilised to display immunogenic peptides of choice directly onto the surface of cells (*Ilca et al., 2018*). We have therefore identified that

manipulation of TAPBPR function may be utilised as a potential immunotherapeutic that facilitates the presentation of both neoantigens and viral-derived peptides, thereby overriding the endogenous cellular antigen processing pathway. Moreover, we have also developed two novel functional assays that enable detailed interrogation of TAPBPR-mediated peptide exchange on MHC I (*Ilca et al., 2018*).

Precisely how tapasin and TAPBPR function at the molecular level remains largely enigmatic. The recently reported crystal structures of human TAPBPR in complex with mouse MHC I captured the endpoint of peptide editing, thereby suggesting that TAPBPR facilitates peptide exchange by widening the peptide binding groove of MHC I at the α2–1 region (*Jiang et al., 2017*; *Thomas and Tampé, 2017*). However, there remains an incomplete understanding of the step-by-step processes by which the two peptide editors, TAPBPR and tapasin, recognise peptide-loaded MHC I molecules and actively facilitate peptide dissociation to result in the conformations observed in the crystal structures. Indeed, McShan et al. have recently used NMR in an attempt to further delineate the dynamic process of peptide exchange on MHC I by TAPBPR (*McShan et al., 2018*).

Intriguingly, the structure reported by Thomas and Tampe identified a loop of TAPBPR that was proposed to interact with the peptide binding groove of MHC I, where the C-terminus of bound peptide usually resides (*Thomas and Tampé, 2017*). While this study demonstrated the localisation of the loop, to date, there is no experimental evidence to support the notion that this loop mediates peptide exchange on MHC I. In contrast, Jiang *et al.* failed to capture the loop in proximity to the peptide-binding groove (*Jiang et al., 2017*), further questioning the relevance and importance of this loop in TAPBPR-mediated peptide exchange. Given the discordance between the data reported for the captured structures and the lack of functional evidence to support any role for this loop, it is vital to reconcile these discrepancies to understand whether the TAPBPR loop is involved in peptide exchange.

Here, we investigate the functional importance of the K22-D35 loop using two newly developed assays in combination with immunopeptidomic analysis. Our data demonstrates that this loop is critical for peptide dissociation from MHC I. Furthermore, we highlight key molecular features governing TAPBPR:MHC I interaction and provide insight into the mechanism(s) of peptide selection on MHC I molecules.

## Results

### The TAPBPR K22-D35 loop lies at the interface with the MHC I peptide binding groove

Prior to the recent determination of the TAPBPR-MHC I crystal structures (*Jiang et al., 2017*; *Thomas and Tampé, 2017*), we docked our model of TAPBPR onto a previously determined structure of HLA-A2, using our mutagenesis data that identified critical regions in the TAPBPR-MHC I interface (*Hermann et al., 2013*). Our docking identified a region of TAPBPR that lies close to the peptide binding groove of MHC I, in the proximity of the F pocket (*Figure 1a*, dotted circle). This region contained a loop that differs between tapasin and TAPBPR. In tapasin, this loop appears to be rather short and is not sufficiently well ordered in the crystal structure to be visible (*Dong et al., 2009*), while in TAPBPR this loop is significantly longer (*Figure 1b*). The two crystal structures of the TAPBPR-MHC I complex support our prediction regarding the arrangement of TAPBPR relative to MHC I, including this loop region being very near the F pocket (*Figure 1c*). However, the position and orientation of the loop in the structures is poorly defined. In the structure from *Jiang et al. (2017)*, this loop is not sufficiently well ordered to be modelled. In the structure from *Thomas and Tampé (2017)*, the loop has been modelled; however, the electron density into which it was built is not well defined: several side chains and even several of the backbone atoms do not fit the electron density well (*Figure 1d*). It is likely that alternative orientations of the loop would also satisfy the crystallographic data presented. Therefore, it is critically important to verify whether there is any functional role of this loop in peptide editing.

### TAPBPR loop mutants are stably expressed and bind MHC I and UGT1

To test whether the K22-D35 loop of TAPBPR was a functionally important region for mediating peptide selection, we first replaced all the residues in the loop with either glycine, alanine or serine, to

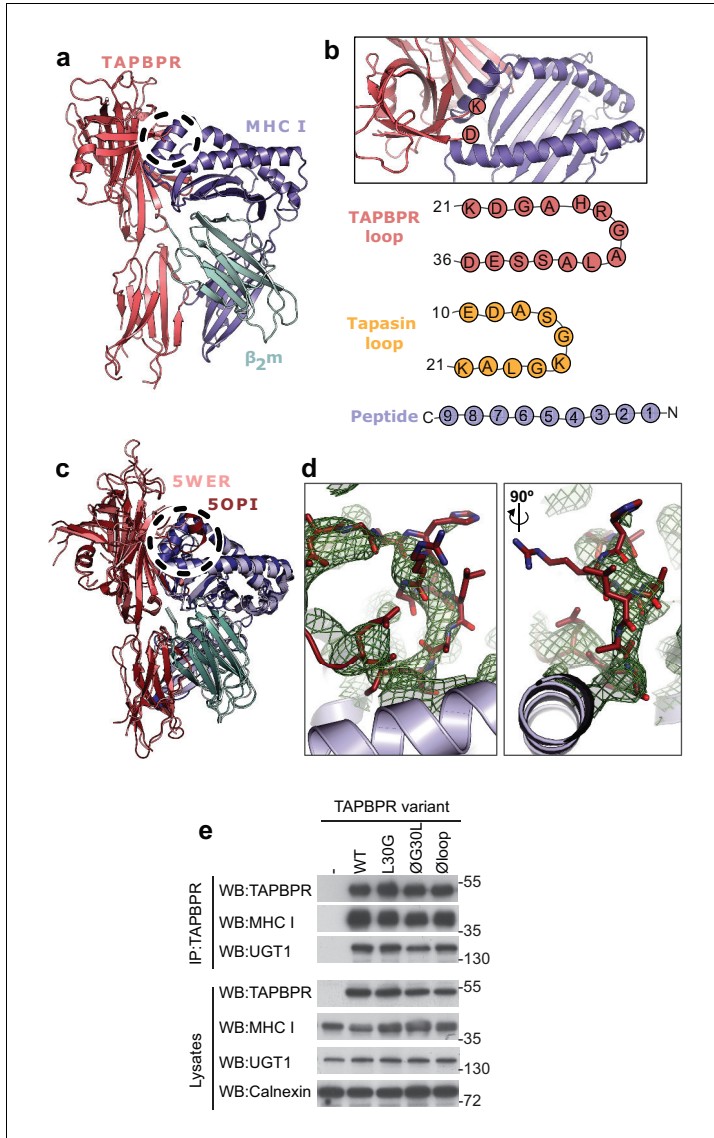

**Figure 1.** TAPBPR loop interactions with MHC I. (**a**) Model of TAPBPR (pink) docked onto MHC I (blue) and β2m (cyan) based on interaction studies (*Hermann et al., 2013*). (**b**) Top panel, illustration of the proximity of the TAPBPR loop region to the peptide binding groove (viewed from the top of complex shown in panel **a**). Lower panel, schematic diagrams of the amino acid composition of the TAPBPR and tapasin loops compared to the length and orientation of a peptide. (**c**) Overlay of two recent X-ray structures of TAPBPR in complex with MHC I (*Jiang et al., 2017*; *Thomas and Tampé, 2017*) (PDB ID 5WER and 5OPI) oriented and coloured to illustrate the similarity to our TAPBPR:MHC I complex (panel **a**). The position of the TAPBPR loop is circled (black dashed line). (**d**) The electron density map ($2F_o$-$F_c$, green mesh) and the built model (maroon sticks, residues D23-E34) are shown for the loop region of TAPBPR (PDB ID 5OPI). Two views of the loop and density are shown rotated by 90 degrees. (**e**) Expression of TAPBPR loop variants in IFNγ treated HeLaM-TAPBPR[KO] and their interaction with MHC I and UGT1. Western blotting for calnexin is included as a loading control. Representative of three independent experiments.

DOI: https://doi.org/10.7554/eLife.40126.002

produce a TAPBPR variant with a potentially functionless loop (TAPBPR[Øloop]) (*Table 1*). More subtle mutations of this loop were designed based on the work by Springer and colleagues, who demonstrated that dipeptides carrying long hydrophobic residues are able to bind to the peptide binding groove of recombinant MHC I and enhance peptide dissociation (*Saini et al., 2015*). Thus, we explored whether a leucine residue at position 30 of the mature TAPBPR protein, the only long

**Table 1.** Panel of TAPBPR loop mutants.
Residues altered are highlighted in red.

| TAPBPR variant | Loop sequence |
|---|---|
| WT | KDGAHRGALASSED |
| Øloop | AAGGSGGGGSGGAA |
| L30G | KDGAHRGAGASSED |
| ØG30L | AAGGSGGGLGGGAA |

DOI: https://doi.org/10.7554/eLife.40126.003

hydrophobic residue within the entire loop, was involved in peptide exchange on MHC I. We created two TAPBPR variants in order to test this. First, we replaced the leucine with glycine in the TAPBPR^WT molecule (TAPBPR^L30G). Second, we reintroduced leucine 30 into TAPBPR with the dysfunctional loop (TAPBPR^ØG30L) (*Table 1*). Upon transduction into TAPBPR-deficient HeLaM cells (HeLaM-TAPBPR^KO), steady state expression of all the TAPBPR loop mutants was similar to TAPBPR^WT and all variants interacted equally well with both MHC I and UGT1 (*Figure 1e*), suggesting the overall protein stability and structure of TAPBPR was not significantly affected by the changes to the loop.

## The K22-D35 loop is essential for mediating peptide exchange on HLA-A*68:02

Recently, we have developed two novel assays which can be used to measure TAPBPR-mediated peptide exchange on MHC I molecules. The first assay established takes advantage of the small proportion of TAPBPR that escapes to the cell surface upon its over-expression in cell lines (*Ilca et al., 2018*). We have demonstrated that plasma membrane expressed wild-type TAPBPR efficiently mediates peptide exchange on cell surface HLA-A*68:02 molecules found on HeLaM cells (*Ilca et al., 2018*). We have also previously shown that TAPBPR-mediated peptide exchange in this assay occurs directly on the cell surface, given that it works on cells incubated at 4°C, which inhibits membrane trafficking (*Ilca et al., 2018*). To initially explore the functional importance of the K22-D35 loop, we tested whether plasma membrane expressed TAPBPR^Øloop was capable of mediating peptide exchange on HLA-A*68:02 to a similar extent as TAPBPR^WT. When exploring the ability of TAPBPR to promote peptide association, we found plasma membrane expressed TAPBPR^Øloop enhanced the binding of an exogenous fluorescent peptide specific for HLA-A*68:02 (YVVPFVAK*V) onto cells to a similar extent as TAPBPR^WT (*Figure 2a*). This may be due, in part, to TAPBPR enhancing the trafficking of associated, peptide-receptive MHC I with it through the secretory pathway. However, when we tested the ability of TAPBPR to mediate peptide dissociation (i.e. removal of the bound fluorescent peptide in the presence of an unlabelled competitor peptide) (*Figure 2b*), in contrast to the efficient exchange observed with TAPBPR^WT, very little, if any, dissociation of fluorescent peptide was observed with TAPBPR^Øloop (*Figure 2c–2e*). This suggests that the K22-D35 loop of TAPBPR is essential for mediating efficient peptide dissociation from HLA-A*68:02.

## L30 is a critical residue of TAPBPR for peptide exchange on HLA-A*68:02

Next, we tested the ability of both plasma membrane bound TAPBPR^L30G and TAPBPR^ØG30L to promote peptide exchange in our assay. As observed above with TAPBPR^Øloop, plasma membrane expressed TAPBPR^L30G and TAPBPR^ØG30L were able to promote the association of exogenous fluorescent peptide onto cells (*Figure 2a*). However, when we explored the ability of TAPBPR^L30G to mediate fluorescent peptide dissociation from HLA-A*68:02, we found that it was incapable of mediating efficient peptide dissociation in the presence of unlabelled competitor (*Figure 2c–2e*). Strikingly, alteration of this single residue had the same effect on the function of TAPBPR as mutating the entire loop. The crucial role of L30 in mediating peptide dissociation was further supported with our observation that TAPBPR^ØG30L, in which the leucine residue alone is restored into the dysfunctional TAPBPR^Øloop molecule, transformed it into a functioning peptide exchange catalyst on HLA-A*68:02, albeit at reduced capability compared to TAPBPR^WT (*Figure 2c–2e*). While TAPBPR^WT

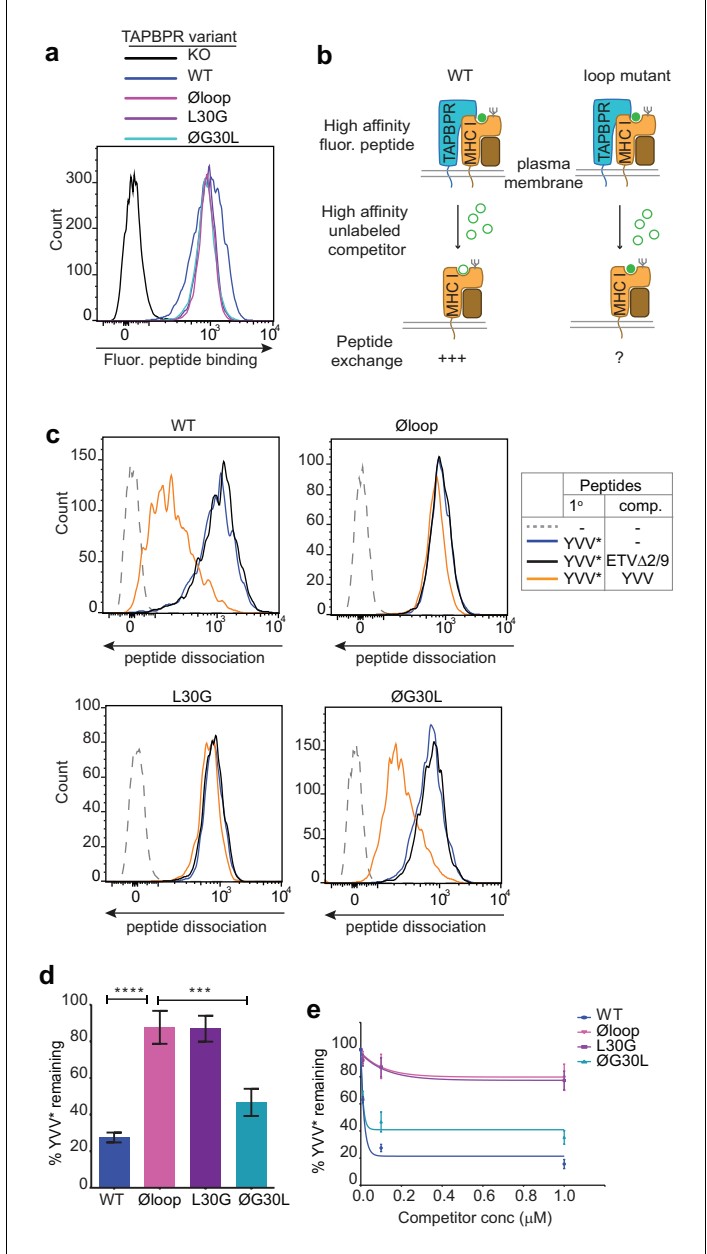

**Figure 2.** The TAPBPR K22-D35 loop is critical for peptide exchange. (a) Typical peptide binding when cells gated for expressing high levels of surface TAPBPR were incubated with 10 nM YVVPFVAK*V peptide for 15 min at 37°C on IFNγ treated HeLaM-TAPBPR$^{KO}$ cells -/+expression of TAPBPR$^{WT}$, TAPBPR$^{Øloop}$, TAPBPR$^{L30G}$ or TAPBPR$^{ØG30L}$. (b) Schematic representation of the experimental workflow used to compare the efficiency of peptide exchange by plasma membrane bound TAPBPR$^{WT}$ with the plasma membrane bound TAPBPR loop mutants. (c) Histograms show the level of dissociation of YVVPFVAK*V (YVV*) in the absence (blue line) and presence of 100 nM non-labelled competitor peptide YVVPFVAKV (YVV)(orange line) or EGVSKQSNG (ETVΔ2/9), a peptide in which the anchors which permit HLA-A*68:02 binding are mutated to produce a non-binding derivative (black line). Similar patterns of dissociation were found on cells incubated at 4°C demonstrating that the peptide exchange occurs directly on the cell surface (see *Figure 2—figure supplement 1*). (d-e) Graphs show the percentage of fluorescent peptide YVVPFVAK*V (YVV*) remaining in the presence of (d) 100 nM or (e) increasing concentrations of the non-labelled competitor peptide YVVPFVAKV as a percentage of the bound YVVPFVAK*V observed in the absence of competitor peptide from four independent experiments. Error bars show -/+SD. ****p≤0.0001, ***p≤0.001 using unpaired two-tailed t-tests.

DOI: https://doi.org/10.7554/eLife.40126.004

*Figure 2 continued on next page*

*Figure 2 continued*

The following figure supplement is available for figure 2:

**Figure supplement 1.** Peptide Exchange at 4°C.

DOI: https://doi.org/10.7554/eLife.40126.005

efficiently promoted rapid peptide exchange at very low concentrations of competitor peptide, both TAPBPR$^{Øloop}$ and TAPBPR$^{L30G}$ were extremely inefficient at mediating peptide exchange on HLA-A*68:02 molecules, even in the presence of high concentrations of competitor peptide (*Figure 2e*). These results suggest that L30 is a critical residue within the loop of TAPBPR for mediating peptide dissociation from HLA-A*68:02.

## Soluble TAPBPR lacking L30 cannot facilitate peptide dissociation from HLA-A*68:02

We have also recently shown that soluble TAPBPR, consisting only of the lumenal domains of the wild-type molecule (i.e. lacking its transmembrane region and cytoplasmic tail) can also efficiently promote peptide exchange on three MHC I molecules: HLA-A*68:02, HLA-A2 and H-2K$^b$ (*Ilca et al., 2018*). Therefore, we can also use soluble TAPBPR as a second means to assay the ability of TAPBPR mutants to mediate efficient peptide exchange on MHC I. Using this approach, the majority of MHC I molecules that TAPBPR will have access to will be folded with bound peptides of relativity high affinity expressed on the surface of cells. The lumenal domains of the TAPBPR variants with C-terminal His-tags were purified using Ni-affinity chromatography from the culture supernatants of transfected 293T cells (*Figure 3a*). Differential scanning fluorimetry revealed all TAPBPR loop variants had a similar melting temperature as TAPBPR$^{WT}$ (*Figure 3b*), indicating the alterations made to the loop had not significantly affected protein folding and stability. Comparison of the ability of the soluble TAPBPR variants to mediate peptide exchange on HLA-A*68:02 molecules on HeLaM cells revealed that TAPBPR$^{WT}$ was most efficient, followed by TAPBPR$^{ØG30L}$, which exhibited ~33% peptide exchange activity relative to TAPBPR$^{WT}$ (in the presence of 10 nM ETVSK*QSNV) (*Figure 3c and d*). However, both TAPBPR$^{L30G}$ and TAPBPR$^{Øloop}$ were unable to efficiently mediating peptide exchange, displaying only ~3% of the exchange activity of TAPBPR$^{WT}$ (in the presence of 10 nM ETVSK*QSNV) (*Figure 3c and d*). As previously shown (*Ilca et al., 2018*), soluble TAPBPR$^{TN5}$, a mutant which cannot bind to MHC I, did not mediate any peptide exchange (*Figure 3c and d*). This hierarchy (WT>ØG30L>L30G>TN5) was maintained over a wide range of exogenous TAPBPR concentrations (*Figure 3e*). Furthermore, the same hierarchical order of peptide exchange efficiency for the variants was observed using another HLA-A*68:02-binding peptide, YVVPFVAK*V (*Figure 3c & d*).

## The K22-D35 loop is essential for soluble TAPBPR to bind peptide-loaded MHC I

When we determined the ability of the soluble TAPBPR variants to bind to HLA-A*68:02, TAPBPR$^{WT}$ was the most efficient binder, followed by TAPBPR$^{ØG30L}$ (*Figure 4a*). However, TAPBPR$^{L30G}$ and TAPBPR$^{Øloop}$ were unable to make productive interactions with surface expressed HLA-A*68:02 (*Figure 4a*). As expected, TAPBPR$^{TN5}$, which has a disrupted binding site for MHC I, was unable to bind to cells (*Hermann et al., 2013*) (*Figure 4a*). As this inability of soluble TAPBPR$^{Øloop}$ and TAPBPR$^{L30G}$ to interact with MHC I contradicted our finding with their membrane-bound counterparts (*Figure 1e*), we determined whether these soluble TAPBPR variants could bind to the total cellular MHC I from cell lysates in pull-down experiments. All soluble TAPBPR variants (with the exception of TAPBPR$^{TN5}$) were capable of binding significant amounts of MHC I, although less MHC I was detected bound to TAPBPR$^{L30G}$ and TAPBPR$^{Øloop}$ (*Figure 4b*). A major difference between surface MHC I on intact cells (*Figure 4a*) and total cellular MHC I in detergent (*Figure 4b*) is the availability of peptide-receptive molecules. Therefore, we speculated that TAPBPR molecules with alterations to the loop are still able to bind to peptide-receptive MHC I but are unable to physically make MHC I peptide-deficient due to their inability to dissociate peptide. Consistent with this, incubation of cells at 26°C, which increases the expression of peptide-receptive MHC I on the plasma membrane (*Ljunggren et al., 1990*; *Schumacher et al., 1990*; *Garstka et al., 2015*), resulted in the

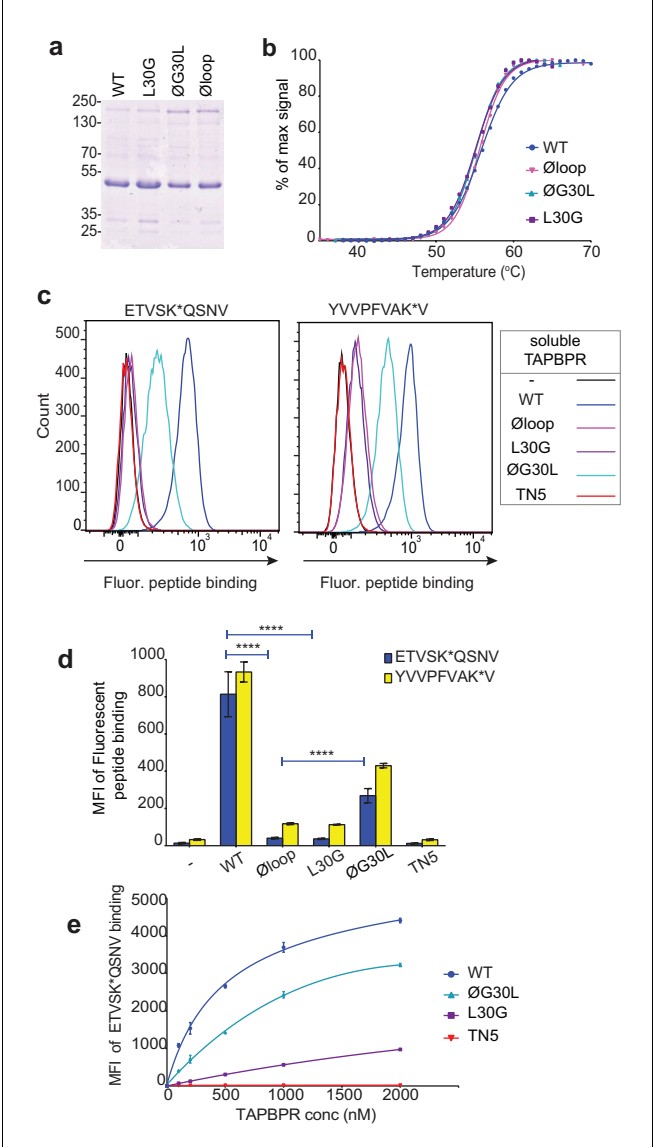

**Figure 3.** Soluble TAPBPR loop variants exhibit reduced ability to mediate peptide exchange on surface HLA-A*68:02 molecules. (a) Expression and purity of soluble forms of WT, L30G, ØG30L, and Øloop TAPBPR variants after their purification from the culture supernatant using Ni-affinity. (b) Differential scanning fluorimetry demonstrates the three TAPBPR loop mutants have equivalent thermal denaturation profiles as TAPBPR$^{WT}$. (c) Histograms of the typical fluorescent peptide binding to IFNγ-treated HeLaM cells incubated -/+100 nM exogenous soluble TAPBPR variant for 15 min at 37°C, followed by incubation with 10 nM ETVSK*QSNV or YVVPFVAK*V for an additional 15 min. TAPBPR$^{TN5}$, in which isoleucine at position 261 is mutated to lysine, to produce a TAPBPR variant which cannot bind to MHC I, is included as a control (d) Bar graphs show the reproducibility of results in (c). (e) Dose response curves of fluorescent peptide binding to IFN-γ treated HeLaM cells incubated with increasing concentrations of the soluble TAPBPR variants prior to the addition of 10 nM ETVSK*QSNV. Error bars represent MFI -/+SD from four independent experiments. ****p≤0.0001 using unpaired two-tailed t-tests.

DOI: https://doi.org/10.7554/eLife.40126.006

TAPBPR loop variants, but not TAPBPR$^{WT}$, exhibiting increased binding to cells (Compare *Figure 4c* with *Figure 4a* summarised in *Figure 4d*). While no significant change was observed in the ability of exogenous TAPBPR$^{WT}$ to bind to cells incubated at the lower temperature, a ~ 7 fold increase in binding of TAPBPR$^{L30G}$ and TAPBPR$^{Øloop}$ was observed at 26°C compared to 37°C, and the binding

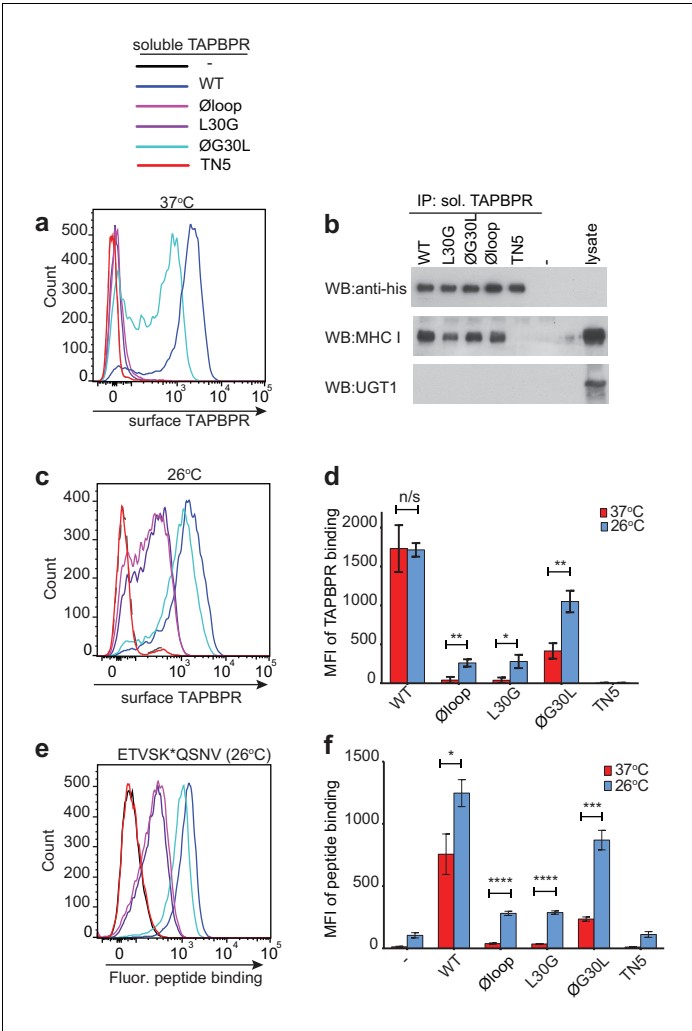

**Figure 4.** Residues K22-D35 are essential for soluble TAPBPR to bind peptide-loaded MHC I. (**a and c**) Histograms of soluble TAPBPR loop variant binding to HeLaM-HLA-ABC^KO cells expressing HLA-A*68:02 incubated with 100 nM TAPBPR at (**a**) 37°C or (**c**) 26°C for 30 min. TAPBPR^TN5, a TAPBPR variant which cannot bind to MHC I, is included as a negative control. (**b**) TAPBPR pull-downs on IFNγ-treated HeLaM-TAPBPR^KO cells incubated with soluble TAPBPR loop mutants reveal all variants are capable of binding to MHC I, but do not bind to UGT1. TAPBPR^TN5 is included as a non-MHC binding control. Data is representative of three independent experiments. (**d**) Bar graph comparing soluble TAPBPR variant binding to HeLaM-HLA-ABC^KO+A*68:02 cells at 37°C with 26°C from three independent experiments. Error bars represent -/+SD. (**e**) Histograms show typical fluorescent peptide binding to IFNγ induced HeLaM cells treated -/+100 nM soluble TAPBPR variants for 15 min at 26°C, followed by incubation with 10 nM ETVSK*QSNV for 15 min at 26°C. (**f**) Bar graph compares ETVSK*QSNV peptide binding to HeLaM cells treated -/+ soluble TAPBPR variants at 37°C with 26°C from three independent experiments. Error bars represent -/+SD. n/s = not significant, *p≤0.05, **p≤0.01, ***p≤0.001, ****p≤0.0001, using unpaired two-tailed t-tests.

DOI: https://doi.org/10.7554/eLife.40126.007

The following figure supplement is available for figure 4:

**Figure supplement 1.** Pre-incubation with high affinity peptide inhibits TAPBPR^Øloop binding to HLA-A*68:02 molecules at 26°C.

DOI: https://doi.org/10.7554/eLife.40126.008

---

of TAPBPR^ØG30L increased by ~2.5 fold (*Figure 4d*). When given access to surface expressed peptide-receptive MHC I upon incubation at 26°C, all soluble TAPBPR variants exhibited a corresponding increase in the ability to mediate peptide exchange (compare *Figure 3c* (37°C) with *Figure 4e* (26°C), summarised in *Figure 4f*). These findings suggest that TAPBPR variants lacking L30 are

indeed able to bind to peptide-receptive MHC I but are unable to physically make MHC I peptide-deficient due to their inability to efficiently facilitate peptide dissociation. Consistent with this, when cells cultured at 26°C were incubated with an HLA-A*68:02-binding peptide prior to testing soluble TAPBPR binding, the peptide significantly reduced the ability of TAPBPR$^{\varnothing loop}$ to bind to cells, while the binding of TAPBPR$^{WT}$ was unaffected (*Figure 4—figure supplement 1*).

## Mutation of the TAPBPR K22-D35 loop alters the peptide repertoire presented on MHC I

Having shown a role for the K22-D35 loop of TAPBPR in mediating efficient peptide dissociation from MHC I, we next determined whether the peptide repertoire presented by MHC I molecules was altered in cells upon mutation of the loop. When we compared the immunopeptidomes of IFNγ-treated HeLaM-TAPBPR$^{KO}$ cells expressing TAPBPR$^{WT}$ with cells expressing TAPBPR$^{\varnothing loop}$, we found significant changes in the peptides presented on MHC I (*Figure 5a*). 461 peptides were found exclusively in the TAPBPR$^{WT}$-expressing cells and 550 peptides were found exclusively in the cells expressing the Øloop variant (*Figure 5a*). Label-free quantitation by mass spectrometry revealed that there were also significant changes in the abundance of some peptides between TAPBPR$^{WT}$ and TAPBPR$^{\varnothing loop}$, with 193 peptides exhibiting increased abundance in cells expressing TAPBPR$^{WT}$ (Red circles, *Figure 5b*) and 222 peptides displaying increased abundance in TAPBPR$^{\varnothing loop}$-expressing cells (Blue circles, *Figure 5b*). These findings demonstrate that there are significant changes in the peptide repertoire presented on MHC I upon mutation of the TAPBPR loop. While there were still large differences in the peptide repertoires between TAPBPR$^{WT}$ and TAPBPR$^{\varnothing G30L}$-expressing cells, based on a presence/absence approach (*Figure 5a*), restoration of the leucine residue into the loop appeared to somewhat reduce some of the changes observed in peptide abundance (*Figure 5b*). Assignment of the identified peptides to the MHC I allotype found in HeLaM cells revealed a similar HLA distribution between the various loop mutants (*Figure 5c and d*).

Given the ability of surface expressed TAPBPR$^{WT}$, but not of TAPBPR$^{\varnothing loop}$, to mediate peptide dissociation from surface MHC I molecules, as identified in *Figure 2*, we performed an additional experiment in which the cells were incubated at 37°C in media after harvesting to permit the surface expressed TAPBPR (~5% of the total TAPBPR pool in these cells [*Ilca et al., 2018*]) to perform peptide dissociation/exchange on surface expressed MHC I molecules, prior to performing immunopeptidomic analysis. While this experiment confirmed the significant changes in the peptide repertoire presented on MHC I upon mutation of the TAPBPR loop (*Figure 5e and f*), it also surprisingly revealed a large effect on the peptides assignable to HLA-A*68:02 between cells expressing TAPBPR$^{WT}$ and TAPBPR$^{\varnothing loop}$ (*Figure 5g and h*). Based on a presence/absence approach, only 29% in TAPBPR$^{WT}$ were now assignable to HLA-A*68:02, compared to 37% for cell expressing TAPBPR$^{\varnothing loop}$-expressing cells (which was similar to the results found when the immunopeptidomics was performed immediately after cell harvesting [*Figure 5d*]). Similarly, abundance analysis revealed >80% of the up-modulated peptides in TAPBPR$^{\varnothing loop}$-expressing cells belonged to HLA-A*68:02, compared to only ~20% of the up-modulated peptides in TAPBPR$^{WT}$ (*Figure 5h*). In keeping with this, we observed similar alterations in HLA-A*68:02-assignable peptides when the leucine residue was restored into the dysfunctional loop (*Figure 5g and h*), as the ones observed for TAPBPR$^{WT}$. This suggests that TAPBPR with a functional loop is preferentially stripping a proportion of peptides from HLA-A*68:02 molecules, but not from the HLA-B or -C molecules found on HeLaM cells. To explore this further, we compared the ability of soluble TAPBPR to bind to soluble heterotrimeric HLA-A*68:02, -B*15:03 or -C*12:03 molecules coupled to beads. This revealed a strong interaction between TAPBPR$^{WT}$ and HLA-A*68:02 (*Figure 5—source data 1*). However, soluble TAPBPR$^{WT}$ failed to bind to heterotrimeric HLA-B*15:03 and –C*12:03 molecules (*Figure 5—source data 1*). This preferential association of TAPBPR with peptide-loaded HLA-A*68:02 over the other MHC I found in HeLaM cells likely explains why we only observe a loss of HLA-A*68:02 peptides in cells expressing TAPBPR with a functional loop. Interestingly, this analysis of soluble HLA molecules coupled on beads also confirmed the considerable reduction in binding of TAPBPR to HLA-A*68:02 upon mutation of the loop (*Figure 5—source data 1*). Taken together, this data demonstrates that the loop is involved in selecting MHC I peptides within cells and highlights the importance of leucine 30 in the peptide selection process, particularly for HLA-A*68:02 molecules.

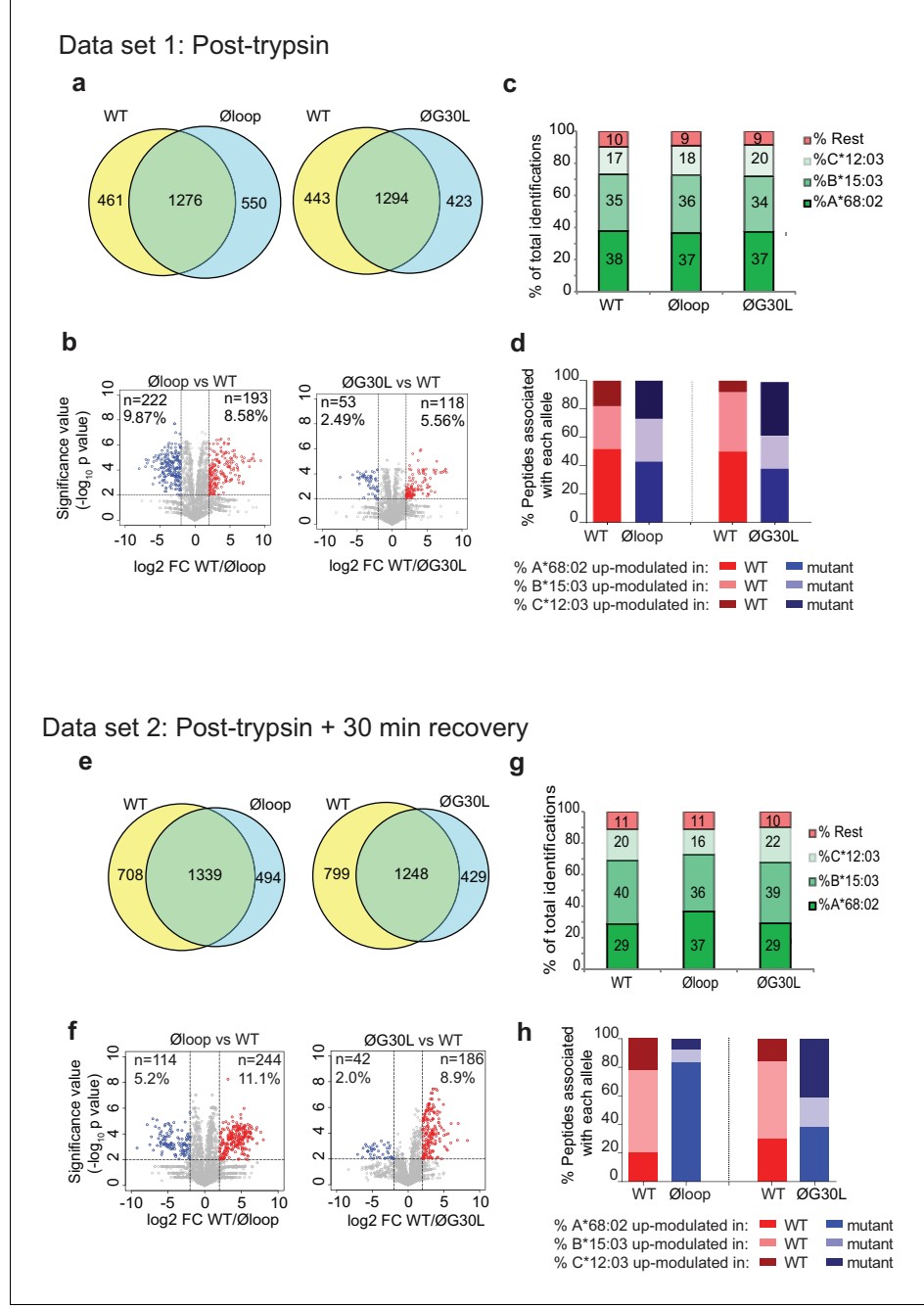

**Figure 5.** Mutation of the K22-D35 loop of TAPBPR changes the peptide repertoire presented on cells. Peptides eluted from W6/32-reactive MHC I complex isolated from IFNγ treated HeLaM-TAPBPR[KO] expressing either TAPBPR[WT], TAPBPR[Øloop] or TAPBPR[ØG30L] were analysed using LC-MS/MS. In dataset 1 (a–d), cells were frozen immediately post-trypsination while in dataset 2 (e–h) cells were allowed to recover in media for 30 min after trypsination, before freezing. The sequences of identified peptides are listed in *Figure 5—source datas 2–7*. The comparison of all five technical replicates for the two datasets is shown in *Figure 5—figure supplement 1*. (a,e) Venn diagrams compare all the identified peptides using a presence/absence approach. (b,f) Volcano plots graphically summarise label-free quantitation, displaying modulated peptides between two cells lines. Colour circles highlight the peptide which are differentially expressed between two cell lines after applying an adjusted p-value of <0.01. The list of these peptides is available in *Figure 5—source datas 8* and *9*. n = number of significantly modulated peptides, % demonstrates the fraction of significantly modulated peptides in a specific cell line compare to all peptides in the comparison. (c,d,g,h) Bar graphs summarise the MHC I molecules (HLA-A*68:02, -B*15:03 or –C*12:03) that the (c,g) identified peptides in a/e, and (d,h) the significantly modulated

*Figure 5 continued on next page*

*Figure 5 continued*

peptides identified in b/f were matched to using the NetMHCpan-4.0. In (c) and (g), peptides not successfully assigned are indicated in orange (rest). Analysis of the peptide repertoire from a further TAPBPR-loop mutant lacking L30 and from a third biological repeat can be found in *Figure 5—figure supplements 2* and *3* respectively. Analysis of the predicted affinity of peptides differential modulated upon mutation of the loop (i.e those in b) and (f) can be found in *Figure 5—figure supplement 4*.

DOI: https://doi.org/10.7554/eLife.40126.009

The following source data and figure supplements are available for figure 5:

**Source data 1.** Binding of TAPBPR to the individual HLA molecules found in HeLaM cells.
DOI: https://doi.org/10.7554/eLife.40126.014

**Source data 2.** Dataset 1 - peptides eluted from W6/32-reactive MHC I complexes from IFNγ treated HeLaM-TAPBPR$^{KO}$ cells expressing TAPBPR$^{WT}$.
DOI: https://doi.org/10.7554/eLife.40126.015

**Source data 3.** Dataset 1 - peptides eluted from W6/32-reactive MHC I complexes from IFNγ treated HeLaM-TAPBPR$^{KO}$ cells expressing TAPBPR$^{∅loop}$.
DOI: https://doi.org/10.7554/eLife.40126.016

**Source data 4.** Dataset 1 - peptides eluted from W6/32-reactive MHC I complexes from IFNγ treated HeLaM-TAPBPR$^{KO}$ cells expressing TAPBPR$^{∅G30L}$.
DOI: https://doi.org/10.7554/eLife.40126.017

**Source data 5.** Dataset 2 - peptides eluted from W6/32-reactive MHC I complexes from IFNγ treated HeLaM-TAPBPR$^{KO}$ cells expressing TAPBPR$^{WT}$.
DOI: https://doi.org/10.7554/eLife.40126.018

**Source data 6.** Dataset 2 - peptides eluted frm W6/32-reactive MHC I complexes from IFNγ treated HeLaM-TAPBPR$^{KO}$ cells expressing TAPBPR$^{∅loop}$.
DOI: https://doi.org/10.7554/eLife.40126.019

**Source data 7.** Dataset 2 - peptides eluted from W6/32-reactive MHC I complexes from IFNγ treated HeLaM-TAPBPR$^{KO}$ cells expressing TAPBPR$^{∅G30L}$.
DOI: https://doi.org/10.7554/eLife.40126.020

**Source data 8.** Dataset 1 - analysis of eluted peptides used to generate volcano plots.
DOI: https://doi.org/10.7554/eLife.40126.021

**Source data 9.** Dataset 2 - analysis of eluted peptides used to generate volcano plots.
DOI: https://doi.org/10.7554/eLife.40126.022

**Source data 10.** Peptides eluted from W6/32-reactive MHC I complexes from IFNγ treated HeLaM-TAPBPR$^{KO}$ cells expressing TAPBPR$^{M29}$.
DOI: https://doi.org/10.7554/eLife.40126.023

**Source data 11.** Dataset 3 - peptide list for third biological repeat for TAPBPR$^{WT}$ expressing cells.
DOI: https://doi.org/10.7554/eLife.40126.024

**Source data 12.** Dataset 3 - peptide list for third biological repeat for TAPBPR$^{∅loop}$ expressing cells.
DOI: https://doi.org/10.7554/eLife.40126.025

**Source data 13.** Dataset 3 - peptides list for third biological repeat for TAPBPR$^{∅G30L}$ expressing cells.
DOI: https://doi.org/10.7554/eLife.40126.026

**Figure supplement 1.** Technical reproducibility of LC-MS/MS measurement.
DOI: https://doi.org/10.7554/eLife.40126.010

**Figure supplement 2.** Mutation of residue A29-D35 in the loop of TAPBPR changes the peptide repertoire presented on cells.
DOI: https://doi.org/10.7554/eLife.40126.011

**Figure supplement 3.** 3$^{rd}$ biological repeat.
DOI: https://doi.org/10.7554/eLife.40126.012

**Figure supplement 4.** Peptide affinity predictions using NetMHC.
DOI: https://doi.org/10.7554/eLife.40126.013

## L30 enables TAPBPR to mediate peptide exchange on MHC I molecules that accommodate hydrophobic amino acids in their F pocket

Given the proximity of the TAPBPR loop to the F pocket of MHC I (*Figure 1*) and that the L30 residue of TAPBPR was both necessary and sufficient for efficient peptide exchange on HLA-A*68:02 (*Figure 2 and 3*), we hypothesised that the TAPBPR loop facilitates peptide dissociation by binding

into the F pocket, thereby competing with the C-terminus of the peptide. If so, this competitive binding would only be possible for MHC I molecules that could accommodate leucine or similar hydrophobic residues in the F pocket. With this in mind, we explored the importance of the TAPBPR K22-D35 loop in mediating peptide exchange on two other MHC I molecules, HLA-A*02:01 and H-2K$^b$, which naturally accommodate hydrophobic amino acids in their F pocket like HLA-A*68:02. Compared to HLA-A*68:02, HLA-A*02:01 accommodates very similar anchor residues in both B and F pockets, whereas H-2K$^b$ has a completely different binding motif, with the exception that it binds a similar anchor residue in the F pocket (*Figure 6a*). Thus, given that both HLA-A*02:01 and H-2K$^b$ could potentially accommodate L30 of the TAPBPR loop in their F-pocket, we predicted TAPBPR-mediated peptide editing on these two additional MHC I would also be dependent on L30.

As observed above with HLA-A*68:02, we found that TAPBPR$^{WT}$ was the most efficient catalyst on both HLA-A*02:01 and H-2K$^b$, followed by TAPBPR$^{∅G30L}$, while both TAPBPR$^{L30G}$ and TAPBPR$^{∅-loop}$ were least efficient (*Figure 6b and c*). For example, the ∅loop, L30G, and ∅G30L variants exhibited ~23%, 32% and 54% activity relative to TAPBPR$^{WT}$, respectively, when measuring NLVPK*VATV binding onto HLA-A2 introduced into HeLaM-HLA-ABC$^{KO}$ cells by transduction (*Figure 6c*). We observed similar trends using another HLA-A2 binding peptide, YLLEK*LWRL, on this cell line, as well as when testing peptide binding to HLA-A2 molecules expressed on MCF-7 cells (*Figure 6—figure supplement 1*). For SIINKEK*L binding to H-2K$^b$ expressed on EL4 cells, the ∅loop, L30G, and ∅G30L variants exhibited ~30%, 31%, 85% activity relative to TAPBPR$^{WT}$, respectively (*Figure 6c*). Thus, although TAPBPR can still mediate peptide exchange on HLA-A2 and H-2K$^b$ in the absence of the loop to some extent, the L30 residue in the K22-D35 loop of human TAPBPR plays a critical role in promoting efficient peptide exchange on HLA-A*68:02, HLA-A2 and H-2K$^b$. The shared dependency on L30 to enable TAPBPR to efficiently mediate peptide exchange on both H-2K$^b$ and HLA-A*68:02 is remarkable, considering the low degree of similarity between these two MHC I, both in their amino acid sequences and in their binding motifs. However, the key shared feature between these MHC I is that both accommodate hydrophobic residues in their F pocket.

## TAPBPR cannot use the loop to efficiently mediate peptide exchange on MHC I molecules with F pocket specificities for non-hydrophobic amino acids

We next tested the role of the TAPBPR loop in catalysing peptide exchange on an MHC I molecule that does not accommodate a hydrophobic residue in its F pocket. In order to only explore the contributing role of the F pocket, with minimal other polymorphisms, we chose HLA-A*68:01, since it only differs from HLA-A*68:02 by five amino acids (*Niu et al., 2013*). Most of the differences between these two HLA are residues dictating the specificity of the F pocket for the C-terminal peptide anchor residue. Hence, while both molecules have a similar anchor residue in their B pocket, the F pocket of HLA-A*68:02 accommodates an aliphatic residue, whereas the F pocket of HLA-A*68:01 accommodates a basic residue (*Figure 6a*) (*Niu et al., 2013*). Strikingly, in contrast to the three MHC I molecules tested previously (which all bind hydrophobic residues in their F pocket), there was a significant impairment in the ability of TAPBPR$^{WT}$ to load peptides onto HLA-A*68:01 (*Figure 6b and c*). Moreover, there was no significant difference in the ability of TAPBPR to exchange peptides on HLA-A*68:01 upon mutation of the loop, with the different loop variants all exhibiting a similar ability as soluble TAPBPR$^{WT}$ to load the fluorescent peptide KTGGPIYK*R onto HLA-A*68:01 expressed on HeLaM-HLA-ABC$^{KO}$ cells (*Figure 6b and c*). Thus, in contrast to HLA-A*68:02, which is extremely receptive to TAPBPR-mediated peptide exchange in a loop-dependent manner, HLA-A*68:01 is significantly less responsive to TAPBPR-mediated peptide exchange, which only occurs in a loop-independent manner. Therefore, the five amino acid differences between these two HLA molecules, three of which are around the F pocket (*Figure 6d*), strongly influence the receptivity of these two MHC I molecules to TAPBPR-mediated peptide exchange, the efficiency of which appears to be strongly influenced by the ability of the K22-D35 loop (specifically L30) to interact with the peptide binding groove. These findings support the concept that the L30 residue of TAPBPR is capable of binding into the F pocket of MHC I in order promote the dissociation of the bound peptide in a competitive manner.

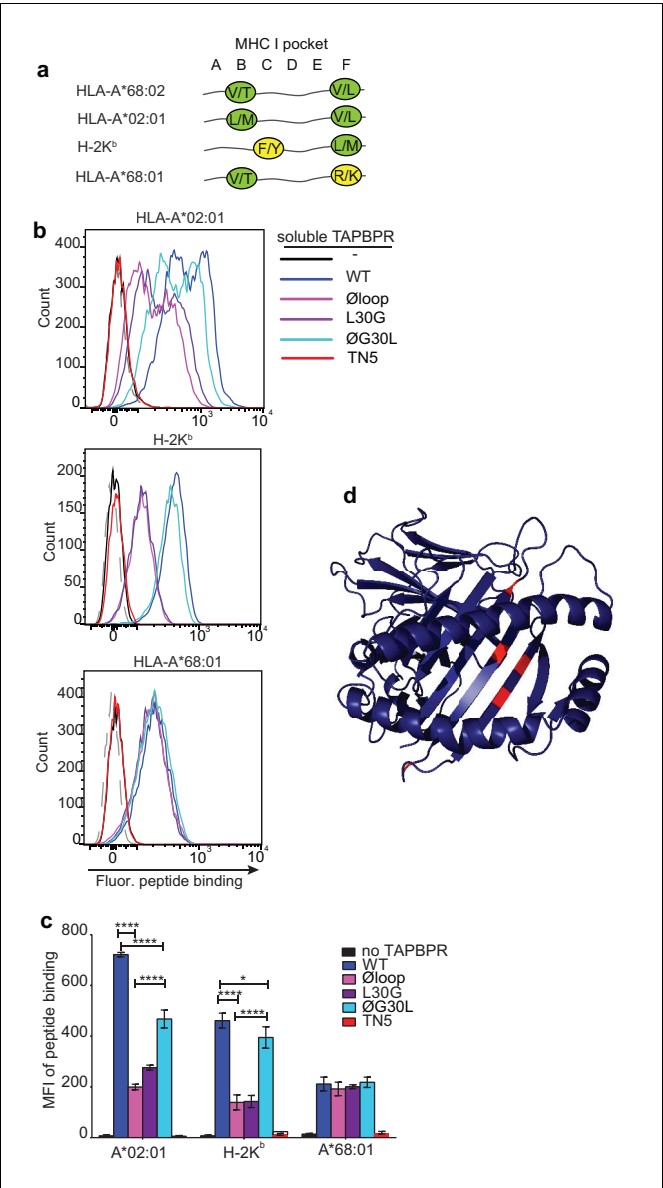

**Figure 6.** F pocket specificity for hydrophobic residues influences the ability of TAPBPR to edit peptides in a loop-dependent manner. (a) Comparison of the A-F pocket specificities of the HLA-A*68:02, HLA-A*02:01, H-2K[b] and HLA-A*68:01 peptide binding grooves. (b) Binding of fluorescent peptide to IFNγ-treated HeLaM-HLA-ABC[KO] transduced with HLA-A*02:01, mouse EL4 cells (which express H-2K[b]) or HeLaM-HLA-ABC[KO] transduced with HLA-A*68:01 -/+1 μM soluble TAPBPR variant for 15 min at 37°C, followed by incubation with 10 nM NLVPK*VATV (HLA-A2 binding peptide) for 60 min, 1 nM SIINFEK*L (H-2K[b] binding peptide) for 30 min or 100 nM KTGGPIYK*R (HLA-A*68:01) for 60 min at 37°C. (c) Bar graph summarising the peptide exchange by soluble TAPBPR variants as performed in b). Error bars represent MFI -/+ SD from four independent experiments. ****p ≤ 0.0001, ***p ≤ 0.001, *p ≤ 0.05, using unpaired two-tailed t-tests. (d) Structure of MHC I from above the binding groove and the different amino acids between HLA-A*68:02 and –A*68:01 highlighted in red.

DOI: https://doi.org/10.7554/eLife.40126.027

The following figure supplement is available for figure 6:

**Figure supplement 1.** Peptide loading by soluble TAPBPR onto HLA-A2 molecules.

DOI: https://doi.org/10.7554/eLife.40126.028

## Mutation of residue 116 in the MHC I F pocket alters TAPBPR binding

Our data thus far is consistent with the L30 residue of the TAPBPR loop binding to the F pocket of MHC I molecules which accommodate hydrophobic residues. One residue that differs between HLA-A*68:02 and -A*68:01 and was reported to be crucial in determining the F pocket specificity of MHC I for peptide residues is the one on position 116 (*Sidney et al., 2008*). The impact of residue 116 on the F pocket architecture is well highlighted in the crystal structures of peptide-bound HLA-A*68:02 and HLA-A*68:01 (*Niu et al., 2013*). Namely, HLA-A*68:02 contains a tyrosine on position 116, whereas HLA-A*68:01 has an aspartate (*Figure 7a*). As captured in the two structures, D116 of HLA-A*68:01 forms strong dipole interactions with both residue R114 of the groove and with the arginine on position 9 of the peptide, determining a strong preference of the HLA-A*68:01 F pocket for basic anchor residues. In contrast, residue Y116 of HLA-A*68:02 does not form any interaction with H114, keeping the hydrophobic patches of the F pocket in an open conformation, which allows it to accommodate hydrophobic peptide residues (*Figure 7a*). We believe that this is the reason why the L30 residue of the TAPBPR loop has access to the F pocket of HLA-A*68:02, but not to the one of HLA-A*68:01.

To explore whether the F pocket specificity of MHC I molecules was indeed crucial for allowing TAPBPR-mediated peptide exchange, we altered the F pocket of both HLA-A*68:02 and -A*68:01, by switching their residues on position 116, producing HLA-A*68:02$^{Y116D}$ and HLA-A*68:01$^{D116Y}$. We subsequently tested their ability to bind soluble TAPBPR. When transduced into HeLaM-HLA-ABC$^{KO}$ cells, the 116-mutated HLA-A68 molecules were expressed at equivalent levels as their WT counterparts (*Figure 7b*). When we tested the ability of soluble TAPBPR$^{WT}$ to bind to cells expressing the HLA-A*68:02 molecules, strikingly, TAPBPR exhibited extremely low levels of binding to HLA-A*68:02$^{Y116D}$ compared to HLA-A*68:02$^{WT}$ molecules (*Figure 7c and d*). Conversely, TAPBPR binding to HLA-A*68:01 was significantly enhanced upon mutation of residue 116 from D to Y (*Figure 7c and d*). The influence of altering the F pocket on TAPBPR binding was further verified on another MHC I molecule, HLA-A2 (*Figure 7—figure supplement 1*). Together, these results demonstrate that the binding of TAPBPR to MHC I molecules is directly influenced by the architecture of the F pocket.

## Mutation of the MHC I F pocket alters TAPBPR-mediated peptide editing

Next, we tested the ability of TAPBPR$^{WT}$ to mediate peptide exchange on the same panel of HLA-A68 molecules. Since HLA-A*68:01 and -A*68:02 have similar residues on position 114, we hypothesized that swapping their 116 residues might also swap their F pocket specificities. Thus, we interrogated the ability of these molecules to bind both the HLA-A*68:02-specific peptide ETVSK*QSNV, as well as the HLA-A*68:01-specific peptide KTGGPIYK*R, in the presence or absence of soluble TAPBPR. While TAPBPR$^{WT}$ efficiently mediated the loading of ETVSK*QSNV onto HLA-A*68:02$^{WT}$ molecules, it failed to load this peptide onto HLA-A*68:02$^{Y116D}$ molecules (*Figure 7e and f*). Similarly, TAPBPR$^{WT}$ failed to load ETVSK*QSNV onto HLA-A*68:01$^{WT}$ molecules, however, efficiently loaded this peptide onto HLA-A*68:01$^{D116Y}$ (*Figure 7e and f*). Moreover, we found that while TAPBPR$^{WT}$ could not load KTGGPIYK*R onto HLA-A*68:02$^{WT}$ molecules, it could efficiently load this peptide onto HLA-A*68:02$^{Y116D}$ (*Figure 7g and h*). Likewise, while HLA-A*68:01$^{WT}$ could bind KTGGPIYK*R in the presence of TAPBPR$^{WT}$, no loading of this peptide was observed onto HLA-A*68:01$^{D116Y}$ (*Figure 7g and h*). These results are in keeping with the fact that the alterations made to the F pocket of the HLA-A68 molecules, and more specifically, to residue 116 alone, have altered peptide specificity. Nonetheless, it is interesting that although low levels of TAPBPR binding are observed to both HLA-A*68:02$^{Y116D}$ and HLA-A*68:01$^{WT}$, TAPBPR is still capable of mediating peptide exchange on these molecule (albeit to a significantly lower extent as compared to their Y116-containing counterparts) and of loading the correct peptide based on their F pocket anchor.

## Discussion

Given the discordance regarding the proximity of the TAPBPR loop in relation to the MHC I peptide binding groove in the recently captured structures (*Jiang et al., 2017*; *Thomas and Tampé, 2017*), and the lack of functional evidence to support any role for this loop, it was vital to directly determine

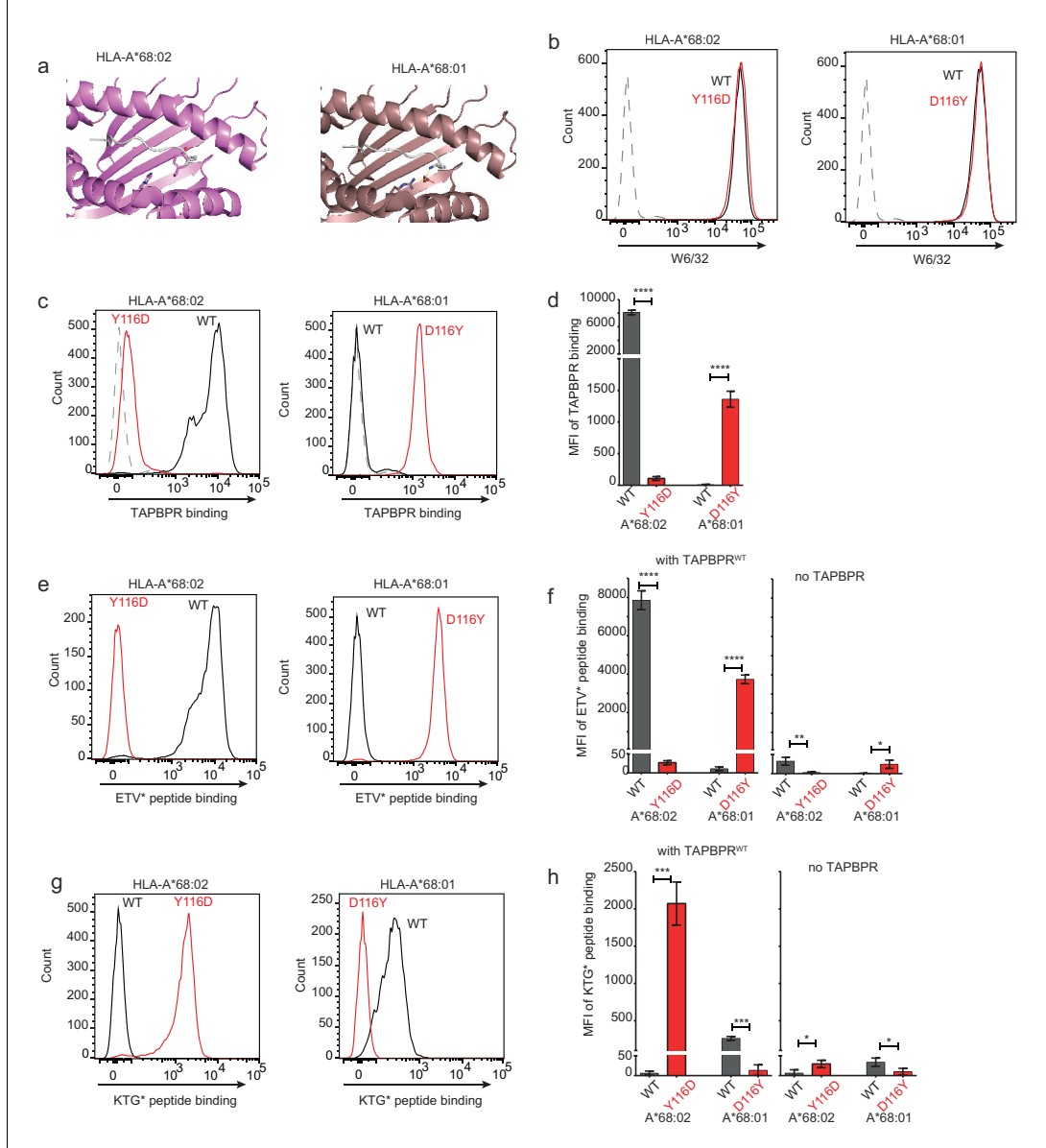

**Figure 7.** Mutation of the MHC I F pocket alters TAPBPR-mediated peptide editing. (a) PyMOL images of the binding grooves of HLA-A*68:02 (PDB ID 4HX1) and -A*68:01 (PDB ID 4HWZ), with residues found at position 116 and 114 highlighted. (b) Histograms show the surface expression of HLA-A*68:02$^{WT}$, -A*68:02$^{Y116D}$, -A*68:01$^{WT}$ and -A*68:01$^{D116Y}$, detected using W6/32, when expressed in HeLa-M-HLA-ABC$^{KO}$ cells. (c, e, g) HeLaM-HLA-ABC$^{KO}$ cells expressing the panel of HLA-A*68 molecules were incubated with 1 μM of soluble TAPBPR for 15 min at 37°C, followed by either (c) detection of surface bound TAPBPR using PeTe4, or incubation with (e) 10 nM ETVSK*QSNV (ETV*) or (g) 100 nM KTGGPIYK*R (KTG*) peptide for 15 min and 60 min, respectively. In (c), staining with an isotype control antibody is included as a control (grey dotted line). While histograms in c, e, g are representative images, the bar graphs in d, f, h summarise the MFI of (d) TAPBPR binding and (f and h) fluorescent peptide binding in the presence and absence of TAPBPR, from three independent experiment ± SD. ****$p \leq 0.0001$, ***$p \leq 0.001$, P** $\leq 0.01$, *$p \leq 0.05$.

DOI: https://doi.org/10.7554/eLife.40126.029

The following figure supplement is available for figure 7:

**Figure supplement 1.** Mutation of the HLA-A2 F pocket alters TAPBPR binding.

DOI: https://doi.org/10.7554/eLife.40126.030

whether this loop contributed at all to peptide exchange. Here, we reveal that upon mutation of the K22-D35 loop, TAPBPR retains binding to MHC I but loses its ability to effectively mediate peptide dissociation. Thus, our work provides compelling evidence that this loop region is critical for TAPBPR to mediate efficient peptide exchange and consequently peptide selection on MHC I molecules. We

establish that the leucine residue in the loop is both necessary and sufficient for TAPBPR to promote peptide dissociation from both human and mouse MHC I molecules, which all typically accommodate a hydrophobic amino acid in their F pocket. Moreover, our findings also demonstrate that the ability of TAPBPR to exchange peptides is severely impaired on MHC I molecules whose F pockets accommodate charged residues. Furthermore, TAPBPR-mediated peptide editing in such scenarios occurs in a loop-independent manner. The differences identified here in the loop-dependency for TAPBPR-mediated editing amongst the various MHC I molecules are striking, especially when comparing HLA-A*68:02 with HLA-A*68:01, which only differ from each other by five amino acids, three of which are located around the F pocket. The extreme differences between these two HLA-A*68 molecules in regard to their susceptibility to TAPBPR-mediated peptide exchange has a remarkable resemblance to the findings reported for HLA-B44 molecules in regard to their dependency on tapasin for optimal peptide selection (*Williams et al., 2002*; *Garstka et al., 2011*). Particularly striking is that residue 116 of HLA I, which severely impacts both the specificity and architecture of the F pocket, seems to play a key role in allowing efficient binding of TAPBPR to HLA molecules.

Although mutation of the loop severely impairs the capacity of TAPBPR to dissociate peptides from MHC I, it seems that TAPBPR is still able, though to a considerably lower extent, to perform this function independently of the loop, across MHC I molecules with different binding motifs. This suggests that there are additional mechanisms by which TAPBPR is capable of mediating peptide exchange on MHC I. These may involve other regions of TAPBPR (for example the jack hairpin [*Thomas and Tampé, 2017*]) and/or could take place according to the recently proposed model of negative allosteric release (*McShan et al., 2018*). Chaperone-mediated peptide editing in such loop-independent scenarios may occur in a more generic and less efficient manner as compared to loop-dependent situations, and may be influenced mainly by the relative stability of individual peptide:MHC I complexes. Thus, one can envisage that TAPBPR can associate with MHC I molecules containing peptides that are intrinsically prone to dissociation (i.e. sub-optimally loaded peptide) in a loop-independent manner.

The functional evidence provided here highlights that the process of peptide editing is complex, multifaceted and is likely to involve dynamic movements of the loop region of TAPBPR as it probes the contents of the MHC I peptide binding groove. Our findings therefore help to explain the ambiguities in the loop conformation identified between the two solved crystal structures. Indeed, our data regarding the critical role of different residues in the MHC I groove to alter peptide exchange efficiency by TAPBPR may implicate an intrinsic difference in the ability of the specific mouse MHC I molecules crystallised with TAPBPR to accommodate hydrophobic residues in their F pocket. This may be of particular importance in the structure where modifications were made to H-2D$^d$ to help stabilise it in a peptide-receptive form (*Jiang et al., 2017*), which may have occluded the ability of the loop to insert into the F pocket.

Our findings are supportive of the following molecular mechanism for peptide exchange by TAPBPR (*Figure 8*). When MHC I enters a state in which the bound peptide partially dissociates (i. e. 'breathes') at the C-terminus from the F pocket, TAPBPR binds to the MHC I groove, at this point in a transient manner, inserting its 22–35 loop into the groove. The leucine 30 residue of the TAPBPR loop binds into the F pocket of the groove, inhibiting the reassociation of the C-terminal anchor residue of the peptide, if hydrophobic, in a competitive manner. The resulting high-free energy intermediate allows TAPBPR to force the MHC I into a conformation to which it can bind more stably, pulling the α2–1 region of the peptide binding groove away from the peptide, as captured in the crystal structures (*Jiang et al., 2017*; *Thomas and Tampé, 2017*). This further prevents the rebinding of the peptide to the groove and thus promotes complete peptide dissociation. Given its stable interaction with peptide-receptive MHC I molecules, TAPBPR prevents these empty MHC I molecules from 'crashing' and we speculate that this consequently allows binding of incoming peptides with affinities for MHC I above the threshold required to outcompete TAPBPR. This final step may share similar features with the tug-of-war model that was previously proposed for tapasin (*Fisette et al., 2016*).

Intriguingly, our data suggests that the loop enables TAPBPR to dissociate peptides with a relatively high affinity for MHC I. This ability of TAPBPR to use its loop to lever peptide out of the MHC I binding groove is consistent with TAPBPR performing peptide editing after tapasin. The identification here of the critical importance of the leucine residue in the TAPBPR loop raises interesting

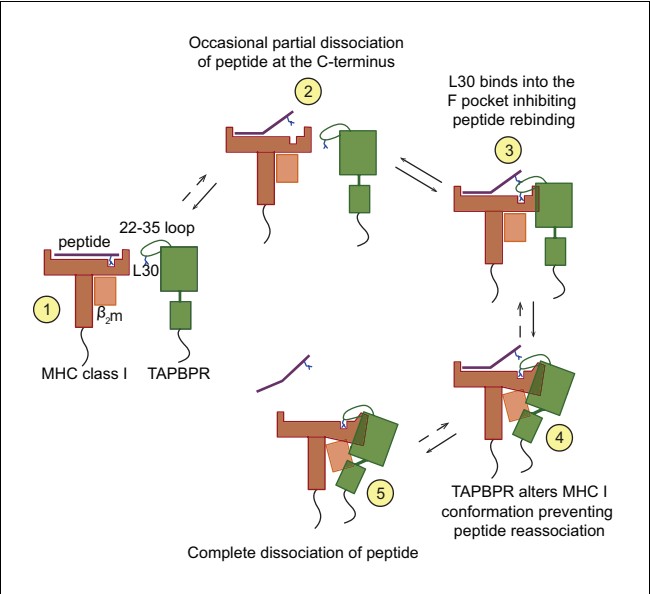

**Figure 8.** Proposed model of the peptide exchange mechanism of TAPBPR on MHC class I.
DOI: https://doi.org/10.7554/eLife.40126.031

questions regarding the different properties of the TAPBPR and tapasin loops and how these mediate step-wise editing to ensure optimal peptide loading of MHC I.

We have recently shown that both plasma membrane-targeted TAPBPR and exogenous soluble TAPBPR can be used to display immunogenic peptides on cell surface MHC I molecules, and, consequently, induce T-cell-mediated killing of target cells (*Ilca et al., 2018*). This observation presents previously unappreciated translational opportunities for utilising TAPBPR as a future immunotherapeutic. For example, TAPBPR could potentially be used to load immunogenic peptides onto tumours to target them from recognition by cytotoxic T cells. Utilising TAPBPR in this manner may promote tumour immunogenicity and increase the clinical efficacy of immune checkpoint inhibitors when used in combination. Here, we have demonstrated the essential role of the TAPBPR loop in the loading of exogenous peptides onto cell surface MHC I molecules and highlight the critical importance of incorporating this structural motif of TAPBPR into compounds developed with a therapeutic intent. Thus, innovative design of the loop may help tailor TAPBPR to specific MHC I molecules and/or alter the properties of the immunogenic peptides loaded by TAPBPR. For example, insight regarding the loop may allow us to tailor TAPBPR to MHC I molecules on which the naturally occurring loop does not work. Furthermore, we may also be able to improve peptide binding on MHC I molecules on which TAPBPR is already able to work on. Consequently, insight into the functional residues of the loop could expand the range of patients that may be responsive to TAPBPR-based therapies.

## Materials and methods

### Docking of TAPBPR with MHC I

Dockings of TAPBPR with MHC I were carried out prior to the recent structure determination of the TAPBPR:MHC I complex structures. For the docking studies, we used our previously determined model for human TAPBPR based on the structure of tapasin (*Hermann et al., 2013*) and human HLA-A2 (*Saper et al., 1991*) (PDB ID 3HLA). TAPBPR was manually docked onto MHC I with the following restraints: (i) the membrane-proximal domains of each component, α3 of MHC I and the IgC domain of TAPBPR were oriented so as to allow these proteins to maintain complex formation when membrane-embedded; (ii) the surface shape complementarity of the components was maintained; (iii) the side chain of residue T134 of MHC I was kept within 10 Å of the sidechains of residues E205-Q209 (TN6 patch) and I261 (TN5 patch) on the IgV domain of TAPBPR; and (iv) sidechains of E222 and D223 of MHC I were kept within 10 Å of the sidechains of residues R335 (TC2) and Q336-S337

(TC3) on the IgC domain of TAPBPR. These last two restraints being mutations known to knock out the interaction in cells (*Hermann et al., 2013*) and the relatively loose restraints allowing for some conformational flexibility of the components upon complex formation.

## Constructs and cell lines

HeLaM cells, a variant HeLa cell line that is more responsive to IFN (*Tiwari et al., 1987*) (a gift from Paul Lehner, University of Cambridge, UK), their modified variants, and HEK-293T (from Paul Lehner, University of Cambridge, UK) were maintained in Dulbecco's Modified Eagle's medium (DMEM; Sigma-Aldrich, UK), supplemented with 10% fetal calf serum (Gibco, Thermo Fisher Scientific), 100 U/ml penicillin and 100 µg/ml streptomycin (Gibco, Thermo Fisher Scientific) at 37°C with 5% $CO_2$. All cells were confirmed to be mycoplasma negative (MycoAlert, Lonza, UK). Authenticity of HeLaM was verified by tissue typing for HLA molecules and by the continuous confirmation that these cell lines had the expected HLA class I tissue type monitored by staining with specific HLA antibodies and by mass spectrometry.

TAPBPR$^{WT}$ and TAPBPR$^{TN5}$ constructs cloned in the lentiviral vector pHRSIN-C56W-UbEM have previously been described (*Boyle et al., 2013*; *Hermann et al., 2013*). TAPBPR$^{Øloop}$, in which amino acids comprising the 22–35 loop were replaced with amino acids glycine, alanine and serine, was created from TAPBPR$^{WT}$pHRSIN-C56W-UbEM using the following procedure: First, amino acids 22–28 were replaced using quick-change site-directed mutagenesis using primers M22-for and M22-rev (see *Table 2* for primer sequences). Subsequently, amino acids 29–35 were replaced using a two-step PCR. For this, the TAPBPR insert was amplified in two separate pieces, starting from each side of the mutation site (primers TAPBPR$^{WT}$-BamHI-for and M29-rev for the N terminus-containing side and primers M29-for and TAPBPR$^{WT}$-NotI-rev for the C terminus-containing side). Subsequently, the two pieces bearing complementary regions over the mutated site were used in a second PCR reaction to amplify the whole TAPBPR mutated insert using primers TAPBPR$^{WT}$-BamHI-for and TAPBPR$^{WT}$-NotI-rev. TAPBPR$^{L30G}$ and TAPBPR$^{ØG30L}$ were generated from TAPBPR$^{WT}$ and TAPBPR$^{Øloop}$ respectively, using primers L30G-for and L30G-rev or ØG30L-for and ØG30L-rev, by quick change site-directed mutagenesis.

The HLA-A*68:02$^{WT}$ construct was cloned by consecutive rounds of quick-change site-directed mutagenesis, using the HLA-A*68:01$^{WT}$ construct as a template (see *Table 3* for primer sequences). Since residue 116 was mutated last in this process, the HLA-A*68:02$^{Y116D}$ mutant was the final intermediate in this cloning process. The HLA-A*68:01$^{D116Y}$ was cloned by quick-change site-directed mutagenesis, using primers A6801-D116Yonly-Fwd and A6801-D116Yonly-Rev. The HLA-A2$^{Y116D}$ mutant was cloned by quick-change site-directed mutagenesis, using primers A2-Y116D-Fwd and A2-Y116D-Rev.

Reconstitution of the TAPBPR variants into the TAPBPR-knockout HeLaM cell line (HeLaM-TAPBPR$^{KO}$), and HLA into the HeLaM-HLA-ABC$^{KO}$ cells was performed using lentiviral transduction and the cells were subsequently cultured as previously described (*Neerincx et al., 2017*;

**Table 2.** Primers used for cloning and generation of TAPBPR loop mutants.

| Primer name | Primer sequence 5'−3' |
| --- | --- |
| TAPBPR$^{WT}$- BamHI-for | GCGCGGATCCAGCAGCCTCCATGGGCACACAGGAGGGC |
| TAPBPR$^{WT}$- NotI-rev | GCGCGCGGCCGCTCAGCTGGGCTGGCTTACA |
| M22-for | GTCCTAGACTGTTTCCTGGTGGCGGCCGGTGGGAGCGGTGGAGCTCTCGCCAGCAGTG |
| M22-rev | CACTGCTGGCGAGAGCTCCACCGCTCCCACCGGCCGCCACCAGGAAACAGTCTAGGAC |
| M29-for | GCGGCCGGTGGGAGCGGTGGAGGTGGCAGCGGCGGTG |
| M29-rev | TCCACCGCTCCCACCGGCCGCCACCAGGAAACAGTCTAGGAC |
| L30G-for | GTGGAGCTGGCGCCAGCAGT |
| L30G-rev | ACTGCTGGCGCCAGCTCCAC |
| ØG30L-for | GGTGGAGGTCTGGGCGGCGGTGC |
| ØG30L-rev | GCACCGCCGCCCAGACCTCCACC |

DOI: https://doi.org/10.7554/eLife.40126.032

**Table 3.** Primers used for generating HLA mutations.

| Primer name | Primer sequence 5′−3′ |
| --- | --- |
| A6801_V12M_Fwd | CTACACTTCCATGTCCCGGC |
| A6801_V12M_Rev | GCCGGGACATGGAAGTGTAG |
| A6801_M97R_Fwd | CACCATCCAGAGGATGTATGGC |
| A6801_M97R_Rev | GCCATACATCCTCTGGATGGTG |
| A6801_S105P_Fwd | CGTGGGGCCGGACGGGC |
| A6801_S105P_Rev | GCCCGTCCGGCCCCACG |
| A6801_R114H_Fwd | GCGGGTACCACCAGGACGCC |
| A6801_R114H_Rev | GGCGTCCTGGTGGTACCCGC |
| A6801_D116Y_Fwd | GTACCACCAGTACGCCTACG |
| A6801_D116Y_Rev | CGTAGGCGTACTGGTGGTAC |
| A6801_D116Yonly_Fwd | GTACCGGCAGTACGCCTAC |
| A6801_D116Yonly_Rev | GTAGGCGTACTGCCGGTAC |
| A2_Y116D_Fwd | GTACCACCAGGACGCCTACG |
| A2_Y116D_Rev | CGTAGGCGTCCTGGTGGTAC |

DOI: https://doi.org/10.7554/eLife.40126.033

*Neerincx and Boyle, 2018*). Cells were induced with 200 U/ml IFN-γ (Peprotech, UK) for 48–72 hr where indicated.

## Expression and purification of TAPBPR protein

To make secreted forms of the TAPBPR loop variants enumerated above, the lumenal domains were cloned into a modified version of the PB-T-PAF vector where the N-terminal Protein A fusion was removed and a C-terminal His$_6$ tag introduced and expressed in 293 T cells using the PiggyBac expression system (*Li et al., 2013*). 48 hr after transfection, cells were transferred for at least 5 days into selection media (DMEM supplemented with 10% FBS, 1% pen/strep, 3 µg/mL puromycin (Inviv-ogen, San Diego, CA) and 700 µg/mL geneticin (Thermo Fisher Scientific, UK). To induce protein expression, cells were harvested and transferred into DMEM supplemented with 5% FBS, 1% pen/strep and 2 µg/mL doxycycline (Sigma-Aldrich, UK). After 5–7 days, the media was collected and TAPBPR was purified using Ni-NTA affinity chromatography. For purity assessment, elution fractions were analysed by SDS-PAGE, followed by Coomassie staining

## Differential scanning fluorimetry (DSF)

Thermofluor experiments were performed in 96-well low profile clear PCR plates for Viia7 cyclers (Axygen). Reactions of 20 µl comprised of 5 µg protein, 1x protein Thermal Shift Dye (Life Technologies) in PBS pH 7.4. The melting curve was performed using a Viia7 thermocycler between 20°C and 95°C in 1°C steps with 20 s equilibration time per step and fluorescence monitored on the ROX channel.

## Isolation of HLA peptides

HLA class I molecules were isolated from HeLaM-TAPBPR$^{KO}$ cells transduced with either TAPBPR$^{WT}$, TAPBPR$^{Øloop}$ or TAPBPR$^{ØG30L}$ using standard immunoaffinity chromatography employing the pan-HLA class I-specific antibody W6/32 (produced in-house), as described previously (*Kowalewski and Stevanović, 2013*). Tissue typing confirmed the HeLaM cells express HLA-A*68:02, -B*15:03 and –C*12:03.

## Analysis of HLA ligands by LC-MS/MS

Isolated HLA peptides were analysed in five technical replicates. Peptide samples were separated by nanoflow high-performance liquid chromatography (RSLCnano, Thermo Fisher Scientific, Waltham, MA) using a 50 µm × 25 cm PepMap rapid separation liquid chromatography column (Thermo Fisher

Scientific) and a gradient ranging from 2.4% to 32.0% acetonitrile over the course of 90 min. Eluting peptides were analyzed in an online-coupled LTQ Orbitrap XL mass spectrometer (Thermo Fisher Scientific) using a top five CID (collision-induced dissociation) fragmentation method.

## Database search and HLA annotation

Spectra were annotated to corresponding peptide sequences by database search of the human proteome as comprised in the Swiss-Prot database (20,279 reviewed protein sequences, September 27[th] 2013) by employing the SEQUEST HT search engine (*Eng et al., 1994*) (University of Washington) integrated into ProteomeDiscoverer 1.4 (Thermo Fisher Scientific). Data processing was performed without enzyme specificity, with peptide length limited to 8–12 amino acids, and methionine oxidation set as dynamic modification. The false discovery rate was calculated by the Percolator algorithm (*Käll et al., 2007*) and set to 5%. HLA annotation was performed using NetMHCpan-4.0 with a percentile rank threshold of 2%.

## Label-free quantitation

We used label-free quantitation (LFQ) as described previously (*Nelde et al., 2018*) to assess the relative HLA ligand abundances between TAPBPR$^{WT}$, TAPBPR$^{\emptyset loop}$ or TAPBPR$^{\emptyset G30L}$ expressing cells. Briefly, relative quantitation of HLA ligands was performed by calculating the area under the curve of the respective precursor extracted ion chromatogram (XIC) using ProteomeDiscoverer 1.4 (Thermo Fisher Scientific). For LFQ analysis the total injected peptide amount of all samples was normalised prior to LC-MS/MS analysis. Volcano plots were computed using an in-house R script (v3.2) and depict pairwise comparisons of the ratios of the mean areas for each individual peptide in the five LFQ-MS runs. Significant modulation was defined by an adjusted p-value of < 0.01 and a fold change of $\geq \log_2 2$ fold change, as calculated by two-tailed t-tests implementing Benjamini-Hochberg correction.

## MHC I-binding peptides

The following fluorescent MHC I-specific peptides were used (K* represents a lysine labelled with 5-carboxytetramethylrhodaime [TAMRA]): ETVSK*QSNV (HLA-A*68:02), YVVPFVAK*V (HLA-A*68:02), NLVPK*VATV (HLA-A*02:01), YLLEK*LWRL (HLA-A*02:01), SIINFEK*L (H-2K$^b$), and KTGGPIYK*R (HLA-A*68:01). The following unlabeled competitor peptides were used: YVVPFVAKV, which exhibits high affinity of HLA-A*68:02 and EGVSEQSNG, a non-binding derived of ETVSEQSNV, obtained by replacing its anchor residues (amino acids on positions 2 and 9) with glycine. All peptides were purchased from Peptide Synthetics, UK.

## Antibodies

TAPBPR was detected using either PeTe4, a mouse monoclonal antibody (mAb) specific for the native conformation of TAPBPR, raised against amino acids 22–406 of human TAPBPR (*Boyle et al., 2013*) that does not cross-react with tapasin (*Hermann et al., 2013*), or ab57411, a mouse mAb raised against amino acids 23–122 of TAPBPR that is reactive to denatured TAPBPR (Abcam, UK). UGT1 was detected using the rabbit mAb ab124879 (Abcam). MHC class I heavy chains were detected using mAb HC10 (*Stam et al., 1986*). Soluble TAPBPR variants were detected using the mouse anti-polyhistidine mAb H1029 (Sigma-Aldrich). A mouse IgG2a isotype control was also used as a control (Sigma-Aldrich).

## Flow cytometry

Following trypsinisation, cells were washed in 1% bovine serum albumin (BSA), dissolved in 1x phosphate-buffered saline (PBS) at 4°C, and then stained for 30 min at 4°C in 1% BSA containing with PeTe4 or with an isotype control antibody. After washing the cells to remove excess unbound antibody, the primary antibodies bound to the cells were detected by incubation at 4°C for 25 min with either goat anti-mouse Alexa-Fluor 647 IgG (Invitrogen Molecular Probes, Thermo Fisher Scientific). After subsequent three rounds of washing, the fluorescence levels were detected using a BD FACScan analyser with Cytek modifications and analysed using FlowJo (FlowJo, LLC, Ashland, OR).

## Peptide binding assay

For peptide binding in the presence of recombinant TAPBPR, the cells were treated with or without the indicated concentration of recombinant TAPBPR (unless otherwise indicated, we used 100 nM for HLA-A*68:02, 1 μM used for HLA-A*02:01, H-2K$^b$ and HLA-A*68:01). After 15 min, the desired TAMRA-labelled peptide was added to the cells and incubated at 37°C (15 min for HLA-A*68:02, 60 min for HLA-A*02:01 and –A*68:01, or 30 min for H-2K$^b$). For experiments performed at 26°C, following cell seeding and IFNγ stimulation at 37°C for 48 hr, cells were transferred at 26°C for another 12 hr to allow for the expression of sub-optimally loaded MHC I molecules at the cell surface. The TAPBPR and peptide binding was then performed as described above, with all incubation steps being performed at 26°C instead of 37°C (with the exception of cell trypinisation which was carried out at 37°C). In cases where the peptide binding was facilitated by over-expressed TAPBPR, the labelled peptide was directly added to the cells, without using recombinant TAPBPR. Following the peptide treatment, the cells were washed three times in 1x PBS and harvested. The level of bound peptide/cell was determined by flow cytometry, using the YelFL1 channel (Cytek).

## Peptide exchange assay

IFNγ-treated HeLaM-TAPBPR$^{KO}$ cell lines reconstituted with the different TAPBPR variants, were treated with 10 nM TAMRA-labelled peptide of interest diluted in opti-MEM for 15 min at 37°C, as described above. Following the binding step, the peptide-containing media was removed, the cells were washed and then treated with media alone or with different concentrations of non-labelled peptide for another 15 min at 37°C. The cells were then washed and harvested and the level of bound peptide per cell was determined by flow cytometry, using the YelFL1 channel (Cytek).

## Immunoprecipitation, gel electrophoresis and western blotting

For TAPBPR immunoprecipitation experiments from cells over-expressing the panel of TAPBPR variants, cells were lysed in 1% triton X-100 (VWR, Radnor, PN), Tris-buffered saline (TBS) (20 mM Tris-HCl, 150 mM NaCl, 2.5 mM CaCl$_2$) supplemented with 10 mM N-ethylmaleimide (NEM), 1 mM phenylmethylsulfonyl fluoride (PMSF) (Sigma-Aldrich), and protease inhibitor cocktail (Roche, UK) for 30 min at 4°C. Nuclei and cell debris were pelleted by centrifugation at 13,000 × g for 15 min and supernatants were collected. Immunoprecipitation was performed with the TAPBPR-specific mAb PeTe4 (*Boyle et al., 2013*) coupled to protein A sepharose (GE Healthcare), at 5 μg antibody per sample, for 2 hr at 4°C with rotation. Following immunoprecipitation, beads were washed thoroughly in 0.1% detergent-TBS to remove.

For pulldown experiments using soluble TAPBPR proteins, IFNγ-stimulated HeLa-TAPBPR$^{KO}$ cells were harvested, lysed and cleared of cell debris as above. In order to remove cellular factors which bind non-specifically to the sepharose beads, the cell lysate was pre-cleared by treatment with protein A sepharose alone, for 30 min at 4°C. Subsequently, the lysate was aliquoted and incubated with 5 μg of the soluble TAPBPR variant for 90 min at 4°C. Immunoprecipitation of soluble TAPBPR was performed using PeTe4 as above. Soluble TAPBPR was detected on western blots with the anti-polyHis primary antibody. Gel electrophoresis and western blot analysis was performed as described in *Neerincx et al. (2017)*.

## Acknowledgements

We are extremely grateful to Gemma Brewin and Sarah Peacock (Tissue Typing Laboratory, Cambridge University Hospitals NHS Foundation Trust) for both the use of and their guidance using the LABScreen single antigen HLA class I beads and the Luminex Fluoroanalyser system. We thank Ben Challis and John Trowsdale from the University of Cambridge for proofreading our manuscript and Nico Trautwein from Eberhard Karls University Tübingen for assistance with Immunopeptidomics.

## Additional information

### Competing interests

F Tudor Ilca, Andreas Neerincx, Louise H Boyle: Some aspects of the work included in this manuscript form part of a recent patent application. Applicant: Cambridge Enterprise Limited. Application number: 1801323.5, Status: Pending. The other authors declare that no competing interests exist.

### Funding

| Funder | Grant reference number | Author |
| --- | --- | --- |
| Wellcome Trust | 104647/Z/14/Z | Andreas Neerincx<br>Louise H Boyle |
| South African Medical Research Council | | Clemens Hermann |
| Royal Society | UF100371 | Janet E Deane |
| Bosch-Forschungsstiftung | | Ana Marcu<br>Stefan Stefvanovic |
| Wellcome Trust | 109076/Z/15/A | Florin Tudor Ilca |

The funders had no role in study design, data collection and interpretation, or the decision to submit the work for publication.

### Author contributions

F Tudor Ilca, Conceptualization, Data curation, Formal analysis, Writing—original draft, Writing—review and editing; Andreas Neerincx, Conceptualization, Resources, Data curation, Supervision; Clemens Hermann, Conceptualization, Formal analysis; Ana Marcu, Data curation, Formal analysis, Writing—review and editing; Stefan Stevanović, Supervision, Funding acquisition, Methodology; Janet E Deane, Conceptualization, Supervision, Methodology, Writing—original draft, Writing—review and editing; Louise H Boyle, Conceptualization, Resources, Formal analysis, Supervision, Funding acquisition, Writing—original draft, Project administration, Writing—review and editing

### Author ORCIDs

F Tudor Ilca ![ORCID] https://orcid.org/0000-0002-6582-8007
Andreas Neerincx ![ORCID] http://orcid.org/0000-0002-6902-5383
Clemens Hermann ![ORCID] http://orcid.org/0000-0002-0009-9501
Ana Marcu ![ORCID] http://orcid.org/0000-0003-0808-8097
Stefan Stevanović ![ORCID] http://orcid.org/0000-0003-1954-7762
Janet E Deane ![ORCID] http://orcid.org/0000-0002-4863-0330
Louise H Boyle ![ORCID] http://orcid.org/0000-0002-3105-6555

### Decision letter and Author response

Decision letter https://doi.org/10.7554/eLife.40126.038
Author response https://doi.org/10.7554/eLife.40126.039

## Additional files

### Supplementary files

• Transparent reporting form
DOI: https://doi.org/10.7554/eLife.40126.034

### Data availability

All data generated or analysed during this study are included in the manuscript and supporting files. Source data files regarding the lists of peptides presented on MHC class I have been provided for Figures 5.

The following dataset was generated:

| Author(s) | Year | Dataset title | Dataset URL | Database and Identifier |
|---|---|---|---|---|
| Ilca FT, Neerincx A, Hermann C, Marcu A, Stevanovic S, Deane JE, Boyle L | 2018 | Data from: TAPBPR mediates peptide dissociation from MHC class I using a leucine lever | https://dx.doi.org/10.5061/dryad.p5k0156 | Dryad, 10.5061/dryad.p5k0156 |

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
