## [Decision Letter]

Thank you for submitting your article "TAPBPR mediates peptide dissociation from MHC class I using a leucine lever" for consideration by *eLife*. Your article has been reviewed by three peer reviewers, one of whom is a member of our Board of Reviewing Editors, and the evaluation has been overseen by Michel Nussenzweig as the Senior Editor. The following individuals involved in review of your submission have agreed to reveal their identity: Nilabh Shastri (Reviewer #2); Peter Van Endert (Reviewer #3).

The reviewers have discussed the reviews with one another and the Reviewing Editor has drafted this decision to help you prepare a revised submission.

Summary:

This manuscript demonstrates the function of a TAPBPR loop in peptide editing of MHC class I proteins. In particular, the authors highlight a hydrophobic residue in this loop as being important in peptide exchange for MHC class I proteins with a hydrophobic F pocket. The data suggest a model in which the TAPBPR loop competes with the peptide side chain for access to the F pocket and thereby favors peptide exchange as well as editing.

Essential revisions:

1) Strengthening of immunopeptidomics data with biological replicates.

2) Examining why TAPBPR has a preference for A:68:02, but not the other studied class I proteins that also have a hydrophobic F pocket.

3) Studying whether TAPBPR indeed mediates peptide exchange at the cell surface in their assays or in an intracellular recycling compartment.

4) Proving that L30 of TAPBPR indeed binds in the F pocket.

5) Studying the biochemistry of MHC class I – TAPBPR interaction with purified soluble proteins.

Reviewer #1:

This is an interesting study that examines the functional significance of a loop of TAPBPR in the editing of MHC class I bound peptides. The authors show that this loop has a major effect on the efficiency of peptide editing. In particular, they implicate a hydrophobic residue (L30) in this loop. Functional data suggest that this hydrophobic residue binds in the F pocket of certain MHC class I proteins and thereby directly competes with the peptide C-terminus.

1) One of the major conclusions is that residue L30 of the TAPBPR loops binds in the F pocket of MHC class I proteins. While the data are consistent with this conclusion, it is important to directly prove this point. This could, for example, be done by working with a HLA-A*68:02 mutant in which only residues of the F pocket are changed.

2) In the mass spec experiment shown in Figure 2, the representation of HLA-A*68:02 peptides is greatly increased in the most severe loop mutant. The opposite result would be expected based on the other functional data. Is this because peptide binding to HLA-B*15:03 and/or HLA-C*12:03 is even more severely affected?

3) All experiments were performed with cell surfaced expressed MHC class I/peptide complexes. However, the major site of peptide editing by TAPBPR is thought to be the ER. It would therefore be useful to assess the effect of TAPBPR with soluble MHC class I/peptide complexes. Soluble TAPBPR is already available.

Reviewer #2:

In this manuscript, Ilca and colleagues provide insights into the molecular mechanism by which TAPBPR mediates exchange of peptides within the MHC I cargo. TAPBPR is a second peptide editor that functions independently of the peptide-loading complex in the endoplasmic reticulum. However the molecular mechanism by which TAPBPR causes exchange of the peptide bound to MHC I is unknown.

Based upon the crystal structure of TAPBPR (Thomas and Tampe, 2011), the authors here analyzed the potential functional significance of a 14 residue (aa 22-35) loop in this molecule. They first replaced all residues in this loop with either glycine, alanine or serine to generate a "functionless" loop (TAPBPRØloop). Additionally they generated two mutants one of WT TAPBPR with the Leu30Gly mutation and another of the TAPBPRØloop in which the Leu residue was reinserted in position 30 (TAPBPR0G30L). DNA constructs encoding each of these molecules when transfected in HeLa cells (lacking endogenous TAPBPR) were similarly expressed and co-immunoprecipitated with both MHC I and the UGT1.

Analysis of the MHC associated peptides by mass spectrometry showed large changes in the peptide repertoire generated in cells expressing WT TAPBPR versus the Øloop mutant as well as the 0G30L mutant versus WT TAPBPR. Interestingly, the changes in the peptide repertoire were particularly marked for HLA*A68 relative to other HLA allotypes. This is an interesting result that suggests that TAPBPR could have selective effects on some but not other MHC I isotypes.

Further, using novel peptide exchange assays developed earlier by this group, the authors showed that the absence of the loop residues (TAPBPRØloop) rendered the molecule non-functional as did the replacement L30G in WT TAPBPR. This loss of peptide exchange activity was substantially although not completely restored by substitution G30L in the TAPBPRØloop mutant.

Altogether, the authors present a nice story with compelling data to support their conclusions. Their model to explaining how TAPBPR causes weakly bound peptides to be dissociated from the MHC I molecule also looks reasonable. However, how the incoming peptide would get into the MHC I groove remains unexplained.

The paper is well written and most of the ideas come through clearly. I would however, have liked the authors to elaborate some more on the potential for therapeutic uses of their findings. As such the ending of the Discussion is rather vague.

The specific sequences identified by mass spectrometry and summarized in Figure 2 should be deposited in a publicly available database.

Reviewer #3:

Ilca and colleagues study the role of a loop extending from the TAPBPR peptide editor into the F pocket of MHC class I molecules. They present clear evidence for a critical and specific role of a Leu in the loop for editing of peptides with hydrophobic C-terminal residues. Globally the data is convincing and represents a significant progress in our understanding of peptide editing by TAPBPR.

However, I am not convinced with respect to the broad validity of the conclusions. Moreover, some problematic technical issues and data interpretations should be addressed.

- The immunopeptidomics data are apparently based on a single experiment with 5 technical replicates. Examining Figure 5—figure supplement 1, only about 57% of peptides are found in 4/5 or 5/5 replicates for WT, and about 38% for G30L. Considering this level of reproducibility for technical replicates, biological replicates can be expected to be even less reproducible. Relatively low reproducibility (which I believe is not unusual for MS data) also raises doubt about the data on relative peptide amounts. Showing at least one example of reproducibility for biological replicates would make the immunopeptidomics data more convincing.

- Together with data shown later in the paper, the peptidomics data are interpreted to indicate a specific effect of TAPBPR on peptides with hydrophobic C-terminals. However, checking the peptide binding motifs of B*15:03 and C*12:03, it turns out that both also bind peptides with hydrophobic C-terminals: Tyr/Phe for B*15:03, Leu/Met/Val/Phe for C*12:03. Is there any evidence, for example from modeling, that TAPBPR acts preferentially on aliphatic not aromatic hydrophobic C-terminals, explaining the opposite effect on B*15:03 (which would exclude many peptides from editing given the high frequency of class I ligands with Tyr/Phe)? Concerning the opposite effect on C*12:03 versus A*68:02, unless the authors have a good explanation for it, I would expect a cautious remark in the Discussion mentioning that the rules remain to be understood and might include other parameters than hydrophobicity.

- The fact that A*68:02, later shown to be particularly susceptible to editing by TAPBPR, acquires many more ligands in its absence could be seen as a bit counter-intuitive – why does A*68:02 present many peptides in larger amounts in the absence of TAPBPR? One explanation would be that these peptides have low affinity and would therefore be chased in its presence. This should be easy enough to check using the NetMHC algorithm used by the authors.

- Although the authors might show that in another paper (PNAS in revision?), the assumption that over-expressed membrane bound or exogenously added soluble TAPBPR acts directly on the plasma membrane would require some evidence. Class I molecules are known to recycle in HeLa cells and TAPBPR might well recycle with them, mediating intracellular peptide exchange. Incubations are performed for 15min at 37°C, sufficient time for recycling. Does soluble TAPBPR have an effect at low temperature?

- The authors conclude that the mutants bind to empty class I, based on an experiment in which they add TAPBPR to Triton lysates of cells. I feel that this conclusion would be more convincing if they used TAP ko cells for this demonstration. TAP ko cells express a majority of poorly loaded or "empty" class I molecules at the surface at 26°C. If the TAPBPR mutants indeed bind only to empty class I, then pre-incubating TAP ko cells at 26°C in the presence of high affinity peptides should abolish their interaction with class I.

---

## [Author Response]

Essential revisions:1) Strengthening of immunopeptidomics data with biological replicates.

We have included data on two additional biological replicates for TAPBPR^WT^ TAPBPR^Øloop^ and TAPBPR^ØloopG30L^. Furthermore, we have provided data for an additional TAPBPR loop mutant in which residues A29-D35 were mutant. The new data provided confirm that there is a significant change in peptide repertoire upon mutation of the loop.

2) Examining why TAPBPR has a preference for A:68:02, but not the other studied class I proteins that also have a hydrophobic F pocket.

We have provided additional data (see Figure 5—source data 1) that demonstrates that TAPBPR binds very strongly to HLA-A*68:02 but does not bind to any significant level to the two other HLA molecules, HLA-B*15:03 and Cw12, expressed in HeLa cells. Therefore, the apparent preference of TAPBPR to influence the peptide repertoire on HLA-A*68:02, and not the two other HLA molecules expressed in HeLa B*15:03 and C*12:03, seems to be related to the intrinsic ability of TAPBPR to bind to HLA molecule, in addition to the specificity of the F pocket for hydrophobic residues

3) Studying whether TAPBPR indeed mediates peptide exchange at the cell surface in their assays or in an intracellular recycling compartment.

We thank you for raising this important point and apologies for not making this obvious in our first draft of the manuscript. We have already extensively investigated this in our now recently published PNAS paper which demonstrates that surprising TAPBPR can mediate peptide even on cells incubated at 4^o^C, which inhibits intracellular recycling. This demonstrates that TAPBPR-mediated peptide exchange occurs directly on the cell surface. Now that the manuscript demonstrating this is published, we have highlighted this key point in the text in this manuscript. We also include similar data at 4^o^C for both TAPBPR-WT and the loop mutants in this manuscript, which can be found in Figure 2—figure supplement 1.

4) Proving that L30 of TAPBPR indeed binds in the F pocket.

We have now investigated this extensively and the data can be found in a new figure (new Figure 7). To test this, we introduced specific mutations to the F pocket of both HLA-A*68:02 and HLA-A*68:01. The data demonstrate that the residue found at position 116 of HLA I molecules is critical for allowing TAPBPR binding. Furthermore, TAPBPR selects the correct peptide corresponding to the F pocket specificity of the MHC I.

5) Studying the biochemistry of MHC class I – TAPBPR interaction with purified soluble proteins.

We have included the ability of soluble WT-TAPBPR and one loop mutant to bind to soluble variants of the three HLA molecules expressed in HeLa (HLA-A*68:02, -B15:03 and –Cw12:03). This reveals that TAPBPR binds extremely well to HLA-A*68:02, but does not bind to HLA-B*15:03 or Cw12:03. The provided data also shows that mutation of the loop severely impairs TAPBPR binding to soluble HLA-A*68:02. This new data can be found in Figure 5—source data 1.

Reviewer #1:This is an interesting study that examines the functional significance of a loop of TAPBPR in the editing of MHC class I bound peptides. The authors show that this loop has a major effect on the efficiency of peptide editing. In particular, they implicate a hydrophobic residue (L30) in this loop. Functional data suggest that this hydrophobic residue binds in the F pocket of certain MHC class I proteins and thereby directly competes with the peptide C-terminus.1) One of the major conclusions is that residue L30 of the TAPBPR loops binds in the F pocket of MHC class I proteins. While the data are consistent with this conclusion, it is important to directly prove this point. This could, for example, be done by working with a HLA-A*68:02 mutant in which only residues of the F pocket are changed.

Many thanks for suggesting this experiment. To explore this, we have compared the ability of TAPBPR to bind to both WT HLA-A*68:02 and HLA-A*68:01 and to their F pocket mutant counterparts (created by altering residue 116). The mutation of residue 116 changed F pocket specificity of both these MHC I molecules and also significantly altered the ability of TAPBPR to bind to them. More specifically, replacing residue 116 in A*68:02 with the one of A*68:01 reduced the ability of TAPBPR to bind and correspondingly, replacing residue 116 in A*68:01 with the one of A*68:02 increased the ability of TAPBPR to bind. This new data can be found in the new Figure 7. We then confirmed the importance of this residue for TAPBPR binding on a different MHC I molecule, HLA-A2. This can be found in figure 7—figure supplement 1. This new data helps highlight the critical importance of the F pocket in TAPBPR binding and peptide exchange.

2) In the mass spec experiment shown in Figure 2, the representation of HLA-A*68:02 peptides is greatly increased in the most severe loop mutant. The opposite result would be expected based on the other functional data. Is this because peptide binding to HLA-B*15:03 and/or HLA-C*12:03 is even more severely affected?

We have now provided evidence that TAPBPR binds extremely well to HLA-A*68:02, but not to HLA-B*15:03 and C*12:03. Thus, the increase in HLA-A*68:02 assignable peptide upon mutation of the loop is unlikely to be due to severe effect on HLA-B15:03 and C*12:03. Upon reflection of both yours and reviewer 3’s comment that the increase representation of HLA-A*68:02 is counter intuitive, we examined precisely how the cell were prepared for immunopeptidomic analysis. For the original dataset, we allowed the HeLaM cells to recover from trypsination by incubating them at 37^o^C in media, before freezing them for shipping for Immunopeptidomic analysis. Given that we have established that the small pool of surface TAPBPR is functional, we wondered whether the observed increase in HLA-A*68:02 assignable peptides was actually due to TAPBPR with a functional loop (i.e. TAPBPR^WT^ and TAPBPR^ØloopG30L^) stripping HLA-A*68:02 peptides from surface expressed MHC I. To explore this, we performed immunopeptidomic analysis on cells immediately after trypsination without the 37^o^C recovery step. While this additional dataset confirmed that mutation of the loop has a significant effect on the peptide repertoire presented, the peptides assignable to HLA-A*68:02 for TAPBPR^WT^ and TAPBPR^ØloopG30L^ were now normalised. Thus, the observed increase in peptides assignable to HLA-A*68:02 for TAPBPR^Øloop^ expressing cells is not due a specific influence of this mutation on MHC I. Rather, in the original dataset (and in an additional biological repeat provided) the observed effect is due to TAPBPR with a functional loop stripping peptides from HLA-A*68:02 molecules. The loss in the ability of TAPBPR to dissociate peptides upon mutation of the loop explains the increased number of peptides assigned to HLA-A*68:02 for the Øloop mutant, as compared to both WT TAPBPR and the ØG30L mutant.

Note: We have rearranged the figure now in our manuscript and the immunopeptidomic data have moved further down the paper and can now be found in Figure 5.

3) All experiments were performed with cell surfaced expressed MHC class I/peptide complexes. However, the major site of peptide editing by TAPBPR is thought to be the ER. It would therefore be useful to assess the effect of TAPBPR with soluble MHC class I/peptide complexes. Soluble TAPBPR is already available.

In an attempt to address this point in the allocated time frame, we have compared the ability of both TAPBPR^WT^ and TAPBPR lacking a function loop to bind to soluble peptide-bound HLA-A*68:02, B*15:03 and C*12:03, coupled to beads. This revealed TAPBPR^WT^ binds strongly to HLA-A*68:02, but not to HLA-B*15:03 or C*12:03. Furthermore, this analysis reveals the decreased ability of TAPBPR to bind to HLA-A*68:02 upon mutation of the loop, which further confirms our other findings in the manuscript.

We agree that our assays test the ability of TAPBPR to mediate peptide exchange in an atypical location. However, the advantage of our systems, rather than using soluble MHC I, which has been refolded in bacteria, is that the MHC I (and also TAPBPR in one of our two assays) is membrane bound and also glycosylated, which reflects the MHC class I found in the ER more closely than using soluble MHC I. While our systems are not perfect and may not truly reflect what occurs in the ER, we feel these assays are superior to using soluble MHC I expressed in bacteria and refolded with a single peptide.

The immunopeptidomic analysis included reflects the effect of mutating the TAPBPR loop more widely on the whole cellular pool of MHC I. 95% of the TAPBPR:MHC I interactions in these experiments occurs in the ER.

Reviewer #2:[…] All together, the authors present a nice story with compelling data to support their conclusions. Their model to explaining how TAPBPR causes weakly bound peptides to be dissociated from the MHC I molecule also looks reasonable. However, how the incoming peptide would get into the MHC I groove remains unexplained.The paper is well written and most of the ideas come through clearly. I would however, have liked the authors to elaborate some more on the potential for therapeutic uses of their findings. As such the ending of the Discussion is rather vague.The specific sequences identified by mass spectrometry and summarized in Figure 2 should be deposited in a publicly available database.

Many thanks for your positive review of our manuscript. In light of your comments we have included some additional text in the Discussion regarding how we envisage incoming peptide may get into the MHC I groove. Furthermore, we have attempted to strengthen the Discussion section regarding how knowledge of the functional importance of the loop prove insightful for therapeutic use. Finally, the specific sequence of the peptides identified by mass spectrometry have been deposited in a publicly available database.

Reviewer #3:Ilca and colleagues study the role of a loop extending from the TAPBPR peptide editor into the F pocket of MHC class I molecules. They present clear evidence for a critical and specific role of a Leu in the loop for editing of peptides with hydrophobic C-terminal residues. Globally the data is convincing and represents a significant progress in our understanding of peptide editing by TAPBPR.However, I am not convinced with respect to the broad validity of the conclusions. Moreover, some problematic technical issues and data interpretations should be addressed.- The immunopeptidomics data are apparently based on a single experiment with 5 technical replicates. Examining Figure 5—figure supplement 1, only about 57% of peptides are found in 4/5 or 5/5 replicates for WT, and about 38% for G30L. Considering this level of reproducibility for technical replicates, biological replicates can be expected to be even less reproducible. Relatively low reproducibility (which I believe is not unusual for MS data) also raises doubt about the data on relative peptide amounts. Showing at least one example of reproducibility for biological replicates would make the immunopeptidomics data more convincing.

We have included data on two additional biological replicates for TAPBPRWT, TAPBPRØloop and TAPBPRØloopG30L. Furthermore, we have provided data for an additional TAPBPR loop mutant in which residues A29-D35 were mutant. The new data provided confirm that there is a significant change in peptide repertoire upon mutation of the loop.

- Together with data shown later in the paper, the peptidomics data are interpreted to indicate a specific effect of TAPBPR on peptides with hydrophobic C-terminals. However, checking the peptide binding motifs of B*15:03 and C*12:03, it turns out that both also bind peptides with hydrophobic C-terminals: Tyr/Phe for B*15:03, Leu/Met/Val/Phe for C*12:03. Is there any evidence, for example from modeling, that TAPBPR acts preferentially on aliphatic not aromatic hydrophobic C-terminals, explaining the opposite effect on B*15:03 (which would exclude many peptides from editing given the high frequency of class I ligands with Tyr/Phe)? Concerning the opposite effect on C*12:03 versus A*68:02, unless the authors have a good explanation for it, I would expect a cautious remark in the Discussion mentioning that the rules remain to be understood and might include other parameters than hydrophobicity.

Note: we have now rearranged the figures and the immunopeptidomic data can be found in Figure 5. We have included data on the ability of TAPBPR to bind to the individual HLA molecules found in HeLa (Figure 5—source data 1). This reveals TAPBPR binds well to HLA-A*68:02 but does not bind to HLA-B*15:03 or C*12:03. Thus, the reason we observe significant effects on HLA-A*68:02 but not the other HLA in HeLa can be explained by the ability of TAPBPR to bind to these individual HLA I molecules. In other words, we claim that while the F pocket specificity for hydrophobic residues is essential for TAPBPR function on MHC I molecules, not all MHC I molecules carrying a hydrophobic F pocket will interact well with TAPBPR.

- The fact that A*68:02, later shown to be particularly susceptible to editing by TAPBPR, acquires many more ligands in its absence could be seen as a bit counter-intuitive – why does A*68:02 present many peptides in larger amounts in the absence of TAPBPR? One explanation would be that these peptides have low affinity and would therefore be chased in its presence. This should be easy enough to check using the NetMHC algorithm used by the authors.

Upon reflection of your comment, we agree that this data is counter-intuitive.

Thus, as mention in our response to reviewer 1, we examined whether incubation of the cells for 30 min at 37^o^C in media after trypsination to allow them to recover was influencing the results for HLA-A*68:02, permitting functional TAPBPR (i.e. TAPBPR^WT^ and TAPBPR^ØloopG30L^) to strip peptides specifically from surface expressed HLA-A*68:02. When immunopeptidomic analysis was performed on cells immediately after trypsination without the 37^o^C recovery step, the peptides assignable to HLA-A*68:02 for TAPBPR^WT^ and TAPBPR^ØloopG30L^ were now normalised. This data can be found in Figure 5. Thus, the loss in the ability of TAPBPR to dissociate peptides upon mutation of the loop explains the increased number of peptides assigned to HLA-A*68:02 for the Øloop mutant, as compared to both WT TAPBPR and the ØG30L mutant.

As suggested, we have performed NetMHC analysis on the peptides. This analysis can be found in Figure 5—figure supplement 4. However, we failed to see any reliable change in affinity of peptides. This is most likely due to limitations in our experimental design which was set up to explore whether there was global effects on mutation of the loop on MHC I peptide presentation, rather than to explore peptide affinity. Thus, in our experimental set up we need to assign peptides to one of the three MHC class I molecules found in HeLaM cells, which requires applying thresholds and cut-offs which will exclude low affinity peptides. In the future the effects of mutating the TAPBPR loop on MHC I peptide affinity could be explored by using cells expressing a single MHC I molecule.

- Although the authors might show that in another paper (PNAS in revision?), the assumption that over-expressed membrane bound or exogenously added soluble TAPBPR acts directly on the plasma membrane would require some evidence. Class I molecules are known to recycle in HeLa cells and TAPBPR might well recycle with them, mediating intracellular peptide exchange. Incubations are performed for 15min at 37°C, sufficient time for recycling. Does soluble TAPBPR have an effect at low temperature?

We have previously explored this in our manuscript recently published in PNAS, which unfortunately was not available to you at the time of your review of this manuscript. We have found that TAPBPR can indeed mediate peptide exchange on cells incubated at 4^o^C, which inhibits membrane trafficking.

We have included some text in the Results to highlight that we have previously shown that TAPBPR edits peptides on cell surface MHC I molecules. Furthermore, we have also included some data to show that we have performed similar experiments at 4^o^C for both TAPBPR-WT as well as the loop mutants. This can be found in the new Figure 2—figure supplement 1.

- The authors conclude that the mutants bind to empty class I, based on an experiment in which they add TAPBPR to Triton lysates of cells. I feel that this conclusion would be more convincing if they used TAP ko cells for this demonstration. TAP ko cells express a majority of poorly loaded or "empty" class I molecules at the surface at 26°C. If the TAPBPR mutants indeed bind only to empty class I, then pre-incubating TAP ko cells at 26°C in the presence of high affinity peptides should abolish their interaction with class I.

We have now included the variant of requested experiment which we believe helps to address the point you have raised here. We did not use T2 cells, instead we used the HeLaM-HLA-ABC^KO^ overexpressing HLA-A*68:02 incubated -/+ high affinity peptide prior to incubation with soluble TAPBPR. Indeed, this revealed that pre-incubation with a high affinity peptide blocked the ability of TAPBPR^ØLoop^ to bind to the cells, but had no significant effect on the ability of TAPBPR^WT^ to bind to cells. This new data can be found in Figure 4—figure supplement 1.